# Punctuated evolution and transitional hybrid network in an ancestral cell cycle of fungi

Edgar M Medina[1,2], Jonathan J Turner[3], Raluca Gordân[2,4], Jan M Skotheim[3], Nicolas E Buchler[1,2]*

[1]Department of Biology, Duke University, Durham, United States; [2]Center for Genomic and Computational Biology, Duke University, Durham, United States; [3]Department of Biology, Stanford University, Stanford, United States; [4]Department of Biostatistics and Bioinformatics, Duke University, Durham, United States

**Abstract** Although cell cycle control is an ancient, conserved, and essential process, some core animal and fungal cell cycle regulators share no more sequence identity than non-homologous proteins. Here, we show that evolution along the fungal lineage was punctuated by the early acquisition and entrainment of the SBF transcription factor through horizontal gene transfer. Cell cycle evolution in the fungal ancestor then proceeded through a hybrid network containing both SBF and its ancestral animal counterpart E2F, which is still maintained in many basal fungi. We hypothesize that a virally-derived SBF may have initially hijacked cell cycle control by activating transcription via the *cis*-regulatory elements targeted by the ancestral cell cycle regulator E2F, much like extant viral oncogenes. Consistent with this hypothesis, we show that SBF can regulate promoters with E2F binding sites in budding yeast.

*For correspondence: nb69@ duke.edu

Competing interests: The authors declare that no competing interests exist.

## Introduction

The networks regulating cell division in yeasts and animals are highly similar in both physiological function and network structure (*Figure 1*) (*Cross et al., 2011*; *Doonan and Kitsios, 2009*). For example, the cell cycle controls proliferation in response to a variety of internal and external signals during the G1 phase, between cell division and DNA replication. These input signals, including cell growth, are integrated into a gradual increase in cyclin dependent kinase (Cdk) activity, which triggers a feedback loop at the basis of the all-or-none irreversible decision to proliferate (*Bertoli et al., 2013*).

Many of the molecular mechanisms underlying G1 regulation are highly conserved. In animal cells, Cyclin D, in complex with either Cdk4 or Cdk6, initiates cell cycle entry by phosphorylating the retinoblastoma protein, pRb. This begins the inactivation of pRb and the concomitant activation of the E2F transcription factors that induce transcription of downstream cyclins E and A, which complete the inhibition of pRb thereby forming a positive feedback loop (*Bertoli et al., 2013*). Similarly, in budding yeast, the G1 cyclin Cln3-Cdk1 complex initiates the transition by phosphorylating and partially inactivating Whi5, an inhibitor of the SBF transcription factor (*Costanzo et al., 2004*; *de Bruin et al., 2004*; *Nasmyth and Dirick, 1991*; *Ogas et al., 1991*; *Sidorova and Breeden, 1993*). This allows for SBF-dependent transcription of the downstream G1 cyclins *CLN1* and *CLN2*, which also inactivate Whi5 to complete a positive feedback loop (*Skotheim et al., 2008*). Thus, both the biochemical function of G1 regulators and their specific targets are highly conserved (*Figure 1*).

Many of the individual proteins performing identical roles are unlikely to be true orthologs, *i.e.*, it cannot be inferred from sequence identity that the proteins evolved from a common ancestral gene.

**eLife digest** Living cells grow and divide with remarkable precision to ensure that their genetic material is faithfully duplicated and distributed equally to the newly formed daughter cells. This precision is achieved through a series of steps known as the cell cycle. The cell cycle is ancient and conserved across all Eukaryotes, including plants, animals and fungi. However, some of the core proteins present in animals and fungi are unrelated. This raises the question as to how a drastic change could have occurred and been tolerated over evolution.

In animals and plants, a protein called E2F controls the expression of genes that are needed to begin the cell cycle. In most fungi, an equivalent protein called SBF performs the same role as E2F, but the two proteins are very different and do not appear to share a common ancestor. This is unexpected given that fungi and animals are more closely related to one another than either is to plants.

Medina et al. searched the genomes of many animals, fungi, plants, algae, and their closest relatives for genes that encoded proteins like E2F and SBF. SBF-like proteins were only found in fungi, yet some fungal groups had cell cycle regulators like those found in animals. Zoosporic fungi, which diverged early from the fungal ancestor, had both SBF- and E2F-like proteins, while many fungi later lost E2F during evolution.

So how did fungi acquire SBF? Medina et al. observed that part of the SBF protein is similar to proteins found in many viruses. The broad distribution of these viral SBF-like proteins suggests that they arose first in viruses, and a fungal ancestor acquired one such protein during a viral infection. As SBF and E2F bind similar DNA sequences, Medina et al. hypothesized that this viral SBF hijacked control of the cell cycle in the fungal ancestor by controlling expression of genes that were originally controlled only by E2F. In support of this idea, experiments showed that many E2F binding sites in modern genes are also SBF binding sites, and that E2F sites can substitute for SBF sites in SBF-controlled genes. Future experiments in zoosporic fungi, which have animal-like and fungal-like features, would provide a glimpse of how a fungal ancestor may have used both SBF and E2F. These experiments may also reveal why most fungi have retained the newer SBF but lost the ancestral and widely conserved E2F protein.

In yeast, a single cyclin-dependent kinase, Cdk1, binds distinct cyclin partners to perform all the functions of three non-orthologous animal Cdks (Cdk2, 4 and 6) during cell cycle entry (*Liu and Kipreos, 2000*). Furthermore, no member of the transcription factor complex SBF-Whi5 exhibits amino acid sequence identity or structural similarity to any member of the E2F-pRb complex (*Cross et al., 2011*; *Hasan et al., 2013*; *Taylor et al., 1997*). Finally, Cdk inhibitors such as Sic1 and p27 play analogous roles in yeast and mammals despite a total lack of sequence identity (*Cross et al., 2011*). Taken together, these examples imply significant evolution of cell cycle regulatory proteins in fungi and/or animals while the network topology remains largely intact. Although identification of network topology is restricted to a few model organisms and is not as broad as sequence analysis, the similar network topology in budding yeast and animals suggests that this feature is more conserved than the constituent regulatory proteins (*Cross et al., 2011*; *Doonan and Kitsios, 2009*).

The shared presence of E2F-pRb within plants (Archaeaplastida) and animal (Metazoa) lineages would suggest that this regulatory complex, rather than the fungal SBF-Whi5 complex, was present in the last eukaryotic common ancestor (*Cao et al., 2010*; *Doonan and Kitsios, 2009*; *Fang et al., 2006*; *Harashima et al., 2013*; *Hallmann, 2009*). The sequence divergence between G1 regulators is surprising because fungi and animals are more closely related to one another than either is to plants. This fungal-metazoan difference raises the question as to where the fungal components came from. Fungal components could either be rapidly evolved ancestral regulators or have a distinct evolutionary history, which would suggest convergent evolution of regulatory networks.

To address this question, we examine conserved and divergent features of eukaryotic cell cycle regulation. In contrast to previous work that considered a protein family of cell cycle regulators in isolation (*Cao et al., 2010*; *2014*; *Eme et al., 2011*; *Gunbin et al., 2011*; *Ma et al., 2013*;

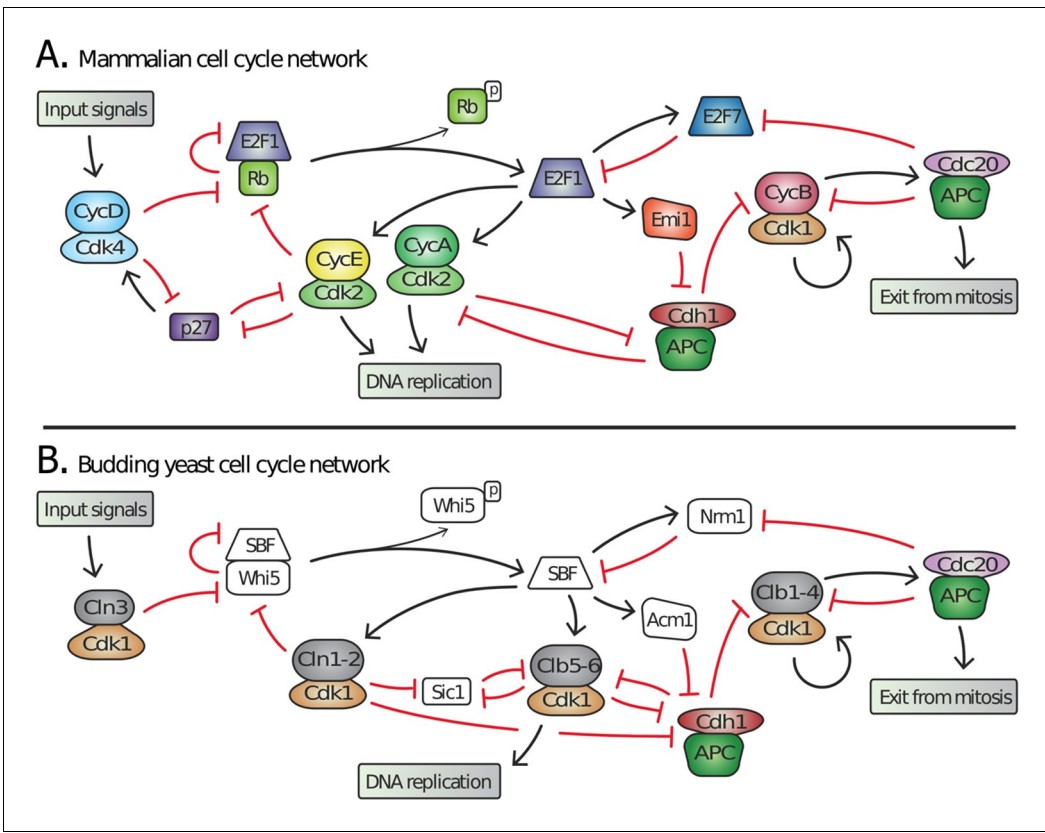

**Figure 1.** Topology of G1/S regulatory network in mammals and budding yeast is conserved, yet many regulators exhibit no detectable sequence homology. Schematic diagram illustrating the extensive similarities between (**A**) animal and (**B**) budding yeast G1/S cell cycle control networks. Similar coloring denotes members of a similar family or sub-family. Fungal components colored white denote proteins with no identifiable animal orthologs.

*Wang et al., 2004*), we studied the evolutionary history of an entire regulatory network across hundreds of species. We examined a greater number of genomes covering most of eukaryotic diversity, including Excavata, Haptophyta, Cryptophyta, SAR (Stramenopiles, Alveolata, Rhizaria), Archaeplastida (plants), Amoebozoa, Apusozoa and the Opisthokonta (animals and fungi). This survey allowed us to estimate the cell cycle repertoire of the last eukaryotic common ancestor (LECA), a prerequisite to clarifying the evolutionary transitions of the cell cycle components of both animals and fungi.

Our results indicate that LECA likely had complex cell cycle regulation involving at least one Cdk, multiple cyclin families, activating and inhibitory E2F transcription factors, and pRb-family pocket proteins. Identifying the LECA repertoire helps establish that the emergence of SBF-Whi5 is abrupt and distinguishes fungi from all other eukaryotes. We show that basal fungi can have both ancestral E2F-pRb and fungal SBF-Whi5 components. Thus, fungal evolution appears to have proceeded through a hybrid network before abruptly losing the ancestral components in the lineage leading to Dikarya. This supports the hypothesis that network structure, rather than the individual components, has been conserved through the transition to fungi and argues against the case of convergent evolution.

Our data confirm that SBF shows homology to KilA-N, a poorly characterized domain present in prokaryotic and eukaryotic DNA viruses. Thus, SBF is not derived from E2F (its functional analog) and likely emerged through horizontal gene transfer in the fungal ancestor. We show that SBF can regulate promoters with E2F binding sites in budding yeast. We then use high-throughput in vitro binding assay data to elucidate the shared nucleotide preferences of E2F and SBF for DNA binding. These data suggest that a viral SBF may have initially hijacked cell cycle control, activating transcription via the *cis*-regulatory elements targeted by the ancestral cell cycle regulator E2F, much like extant viral oncogenes.

## Results

### Reconstruction of complex cell cycle control in the ancestral eukaryote

Recent work shows that the last eukaryotic common ancestor (LECA) already had a complex repertoire of protein families (*Dacks and Field, 2007*; *Eichinger et al., 2005*; *Merchant et al., 2007*). Indeed, all sequenced eukaryotic lineages have lost entire gene families that were present in LECA (*Fritz-Laylin et al., 2010*). In contrast to the growing consensus that LECA had an extensive repertoire of proteins, the prevailing view of the cell cycle in LECA is that it was based on a simple oscillator constructed with relatively few components (*Coudreuse and Nurse, 2010*; *Nasmyth, 1995*). According to the 'simple' LECA cell cycle model, an ancestral oscillation in Cyclin B-Cdk1 activity drove periodic DNA replication and DNA segregation, while other aspects of cell cycle regulation, such as G1 control, may have subsequently evolved in specific lineages. The model was motivated by the fact that Cdk activity of a single Cyclin B is sufficient to drive embryonic cell cycles in frogs (*Murray and Kirschner, 1989*) and fission yeast (*Stern and Nurse, 1996*), and that many yeast G1 regulators have no eukaryotic orthologs (*Figure 1*).

To determine the complexity of LECA cell cycle regulation, we examined hundreds of diverse eukaryotic genomes. We first built sensitive profile Hidden Markov Models (*Eddy, 2011*) for each of the gene families of cell cycle regulators from model organisms *Arabidopsis thaliana*, *Homo sapiens*, *Schizosaccharomyces pombe*, and *Saccharomyces cerevisiae*. These HMMs were then used to query the sequenced eukaryotic genomes for homologs of both fungal and animal cell cycle regulators (see Materials and methods and *Figure 2—figure supplement 1* for a complete list of regulatory families in each genome). Phylogenetic analyses were performed on the detected homologs for accurate sub-family assignment of the regulators and inference of their evolutionary history (see Materials and methods). If LECA regulation were simple, we would expect little conservation beyond the Cyclin B-Cdk1 mitotic regulatory module. However, if LECA regulation were more complex, we would expect to see broad conservation of a wider variety of regulators.

While we did not find either of the fungal regulators (SBF and Whi5) outside of Fungi, we did find animal-like cell cycle regulators in Archaeplastida, Amoebozoa, SAR, Haptophyta, Cryptophyta, Excavata and Metazoa (*Figure 2*). For example, the cyclin sub-families (A, B, D, and E) known to regulate the cell cycle in metazoans (for cyclin phylogeny see *Figure 2—figure supplement 2*) are found across the major branches of eukaryotes. We also found examples of all three sub-families of E2F transcription factors (E2F1-6, DP, E2F7/8) and the pRb family of pocket proteins (for E2F/DP and pRb phylogeny see *Figure 2—figure supplement 3* and *Figure 2—figure supplement 4*). Nearly all species contain the APC specificity subunits Cdc20 and Cdh1/Fzr1, which regulate exit from mitosis and maintain low Cdk activity in G1 (for Cdc20-family APC phylogeny see *Figure 2—figure supplement 5*). Taken together, these data indicate that LECA cell cycle regulation was based on multiple cyclin families, as well as regulation by the APC complex and members of the pRb and E2F families. More broadly, our phylogenetic analyses tend to place the fungal regulators as sister groups to the metazoan regulators, as would be expected from the known eukaryotic species tree. These phylogenies are in agreement with the hypothesis that many fungal and metazoan regulators were vertically inherited from an opisthokont ancestor rather than loss of these regulators in fungi followed by secondary acquisition through horizontal gene transfer.

Members of the Cdk1-3 family (i.e. CdkA in plants) are also broadly conserved across eukaryotes, suggesting they were the primary LECA cell cycle Cdks (for CDK phylogeny see *Figure 2—figure supplement 6*). Other cell cycle Cdk families in animals (Cdk4/6) and plants (CdkB) are thought to be specific to those lineages. However, we found CdkB in Stramenopiles, which may have arrived via horizontal transfer during an ancient secondary endosymbiosis with algae as previously suggested (*Cavalier-Smith, 1999*). We excluded from our analysis other families of cyclin-Cdks, *e.g.*, Cdk7-9, which regulate transcription and RNA processing, and Cdk5 (yeast Pho85), which regulate cell polarity, nutrient regulation, and contribute to cell cycle regulation in yeast (*Cao et al., 2014*; *Guo and Stiller, 2004*; *Ma et al., 2013*; *Moffat and Andrews, 2004*). While interesting and important, an extensive examination of these cyclin-Cdk families is beyond the scope of this work.

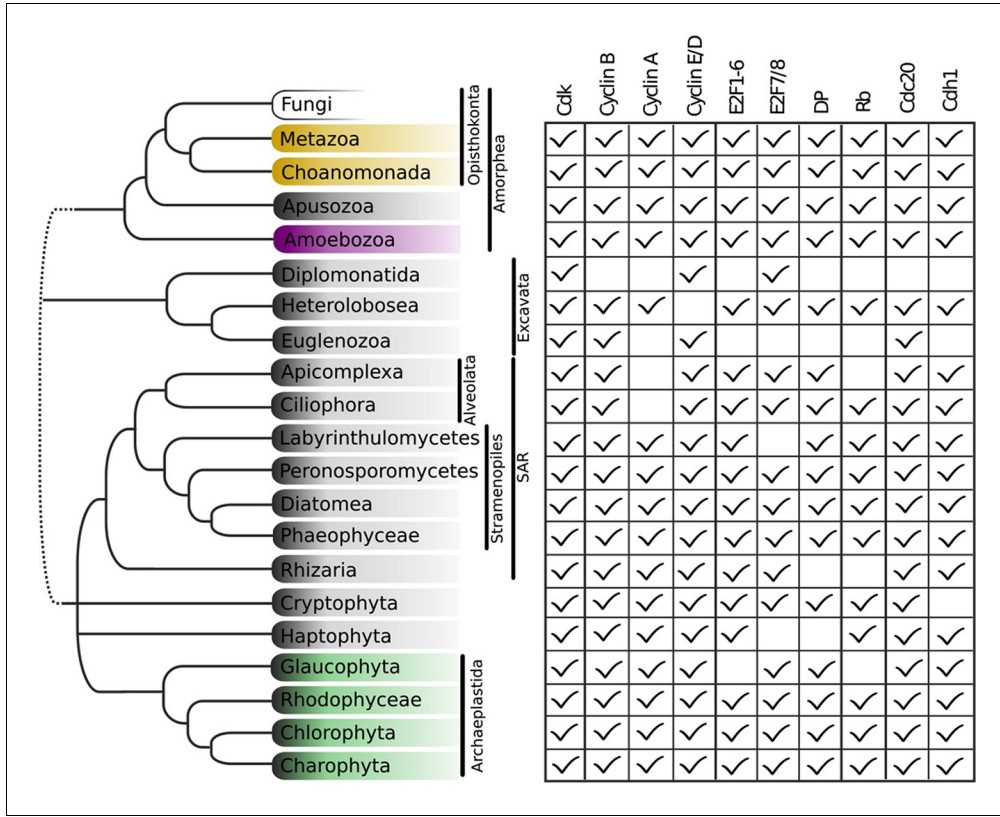

**Figure 2.** Animal and plant G1/S regulatory network components were present in the last eukaryotic common ancestor. Distribution of cell cycle regulators across the eukaryotic species tree (**Adl et al., 2012**). Animals (Metazoa) and yeasts (Fungi) are sister groups (Opisthokonta), and are distantly related to plants (Charophyta), which are members of the Archaeplastida. Check marks indicate the presence of at least one member of a protein family in at least one sequenced species from the corresponding group. We developed profile-HMMs to detect cell division cycle regulators in eukaryotic genomes. For each cell cycle regulatory family (*e.g.*, Cyclins), we used molecular phylogeny to classify eukaryotic sequences into sub-families (*e.g.*, Cyclin B, Cyclin A, Cyclin E/D). See *Figure 2—figure supplement 1* for complete list of regulatory families in all eukaryotic species, and *Figure 2— figure supplement 2* (Cyclin), *Figure 2—figure supplement 3* (E2F/DP), *Figure 2—figure supplement 4* (pRb), *Figure 2—figure supplement 5* (Cdc20-family), and *Figure 2—figure supplement 6* (CDK) for final phylogenies.

The following source data and figure supplements are available for figure 2:

**Source data 1.** Reduced set of eukaryotic cell cycle cyclins for phylogenetic analysis.

**Source data 2.** Complete set of eukaryotic E2F/DP transcription factors for phylogenetic analysis.

**Source data 3.** Complete set of eukaryotic Rb inhibitors for phylogenetic analysis.

**Source data 4.** Reduced set of eukaryotic Cdc20-family APC regulators for phylogenetic analysis.

**Source data 5.** Reduced set of eukaryotic cyclin-dependent kinases for phylogenetic analysis.

**Figure supplement 1.** Comparative genomic data of G1/S regulators across eukaryotes.

**Figure supplement 2.** Reduced phylogeny of eukaryotic cell cycle cyclins.

**Figure supplement 3.** Phylogeny of eukaryotic E2F/DP transcription factors.

**Figure supplement 4.** Phylogeny of eukaryotic Rb inhibitors.

*Figure 2 continued on next page*

*Figure 2 continued*

**Figure supplement 5.** Reduced phylogeny of eukaryotic Cdc20-family APC regulators.
**Figure supplement 6.** Reduced phylogeny of eukaryotic cyclin-dependent kinases.

## A hybrid E2F-pRb-SBF-Whi5 network on the path of fungal evolution

To identify the possible origins of fungal SBF and Whi5, we developed SBF and Whi5 HMMs to query other eukaryotic genomes for homologs of these fungal-specific regulators. We were unable to find any eukaryotic homologs of SBF or Whi5 outside of fungi, with a few exceptions related to DNA viruses that we discuss later. SBF is an important member of a larger family of winged helix-turn-helix transcription factors that includes Xbp1, Bqt4, and the APSES family (Acm1, Phd1, Sok2, Egf1, StuA); see Materials and methods and *Figure 3—figure supplement 1* for a complete list of homologs in each fungal genome. The emergence of the new fungal regulators SBF, which includes the large APSES family, and Whi5 is abrupt and occurs near the split of basal fungi from metazoans (*Figure 3*). The precise location remains unclear because we have only 1 Nuclearid genome (*Fonticula alba*) and because Microsporidia are fast-evolving fungal parasites with reduced genomes (*Cuomo et al., 2012*). Interestingly, the new regulators (SBF and Whi5) and ancestral regulators (E2F and Rb) co-exist broadly across basal fungi and the lineages formerly known as 'zygomycetes' (*Figure 3*). Zoosporic, basal fungi such as Chytrids (*e.g., Spizellomyces punctatus*) can have both fungal and animal cell cycle regulators, which likely represents the ancestral fungal hybrid network. SBF-Whi5 in budding yeast plays a similar role to E2F-pRb in animals, which suggests that these pathways were functionally redundant in an ancestral hybrid network. This redundancy would lead to the evolutionary instability of the hybrid network and could explain why different constellations of components are present in the extant zygomycetes and basal fungi (*Figure 3—figure supplement 1*). For example, the zygomycetes have lost pRb while retaining E2F, which was then abruptly lost in the transition to Dikarya. However, with the possible exception of Microsporidia, all fungi have retained SBF and never completely reverted back to the original ancestral state.

## The SBF and E2F family of transcription factors are unlikely to be orthologs

A simple scenario to explain the emergence of a hybrid network would be gene duplication of the E2F pathway followed by rapid sequence evolution to create a partially redundant SBF pathway. To this end, we scrutinized the highly conserved E2F and SBF DNA-binding domains to detect any sequence and structural homology. We used the Pfam HMMER model of the E2F/DP DNA-binding domain and SBF+APSES DNA-binding domain (KilA-N.hmm), which is homologous to the KilA-N domain (*Iyer et al., 2002*). Our rationale for using KilA-N.hmm from Pfam for remote homology detection of SBF+APSES stems from the fact that it was trained on a diverse set of KilA DNA-binding domains across bacterial DNA viruses, eukaryotic DNA viruses, and fungal SBF+APSES proteins. Thus, it should be a more sensitive HMM model to detect remote KilA-N homologues in other eukaryotic genomes. Our controls (*H. sapiens* genome, *Figure 4A* and *S. cerevisiae* genome, *Figure 4B*) demonstrate that E2F_TDP.hmm is specific to E2F/DP and that KilA-N.hmm is specific to SBF+APSES. We show that genomes with hybrid network (*S. punctatus*, *Figure 4C*, and other basal fungi with both transcription factors, *Figure 4D*) have both E2F/DP and SBF+APSES. E2F_TDP.hmm never hits an SBF+APSES transcription factor and KilA-N.hmm never hits an E2F transcription factor (i.e. there are no scores on the diagonal of the panels in *Figure 4*). Thus, there is no misclassification by the Pfam HMM models. These data suggest that SBF and other KilA-N domains have no more sequence identity to E2F than non-homologous proteins. We find that non-fungal genomes only hit E2F/DP (*Figure 4E*) with the notable exception of *Trichomonas vaginalis*, the only non-fungal genome with E2F/DP and KilA-N homologs (*Figure 4F*). We will discuss the case of *T. vaginalis* in the next section.

To further test the possibility that SBF was a gene duplication of E2F/DP and evolution was so rapid that sequence identity was lost, but structural and functional homology to E2F/DP was maintained, we looked for possible evidence of structural homology. The DNA-binding domains of SBF/MBF (*Taylor et al., 1997*; *Xu et al., 1997*) and E2F/DP (*Zheng et al., 1999*) are structurally classified

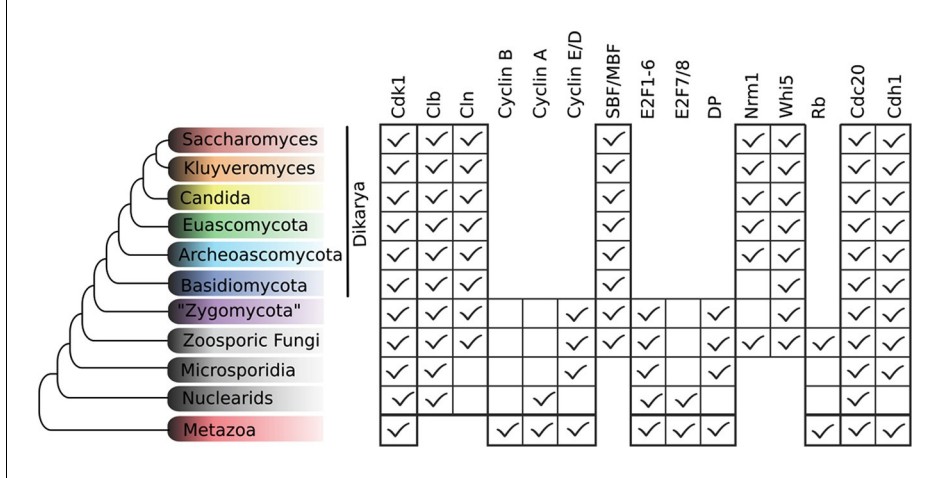

**Figure 3.** Fungal ancestor evolved novel G1/S regulators, which eventually replaced ancestral cyclins, transcription factors, and inhibitors in Dikarya. Basal fungi and 'Zygomycota' contain hybrid networks comprised of both ancestral and fungal specific cell cycle regulators. Check marks indicate the presence of at least one member of a protein family in at least one sequenced species from the group; see *Figure 3—figure supplement 1* for a complete list of homologs in all fungal species. Cells are omitted (rather than left unchecked) when a family is completely absent from a clade. For each fungal regulatory family (*e.g.*, SBF/MBF), we used molecular phylogeny to classify eukaryotic sequences into sub-families (*e.g.*, SBF/MBF, APSES, Xbp1, Bqt4). See Materials and methods for details and *Figure 3—figure supplement 2* (SBF/MBF only), *Figure 3—figure supplement 3* (SBF/MBF +APSES), and *Figure 3—figure supplement 4* (Whi5/Nrm1) for final phylogenies.

The following source data and figure supplements are available for figure 3:

**Source data 1.** Complete set of fungal SBF/MBF transcription factors for phylogenetic analysis.

**Source data 2.** Complete set of fungal SBF/MBF and APSES transcription factors for phylogenetic analysis.

**Source data 3.** Complete set of fungal Whi5/Nrm1 inhibitors for phylogenetic analysis.

**Figure supplement 1.** Comparative genomic data of G1/S regulators across fungi.

**Figure supplement 2.** Phylogeny of fungal SBF/MBF transcription factors.

**Figure supplement 3.** Phylogeny of fungal SBF/MBF and APSES transcription factors.

**Figure supplement 4.** Phylogeny of fungal Whi5/Nrm1 inhibitors.

as members of the winged-helix-turn-helix (wHTH) family, which is found in both prokaryotes and eukaryotes (*Aravind and Koonin, 1999*; *Aravind et al., 2005*; *Gajiwala and Burley, 2000*). Although the DNA-binding domains of E2F/DP and SBF/MBF are both classified as wHTH proteins, they show important differences in overall structure and mode of protein-DNA complex formation that lead us to conclude that it is highly unlikely that they are orthologs.

Many wHTH transcription factors, including the E2F/DP family, have a 'recognition helix' that interacts with the major or minor grooves of the DNA. The E2F/DP family has an RRXYD DNA-recognition motif in its helix that is invariant within the E2F/DP family and is responsible for interacting with the conserved, core GCGC motif (*Zheng et al., 1999*) (see *Figure 5A*: red structure). The RRXYD recognition motif is strikingly conserved in E2F/DP across all eukaryotes, including the E2F/DP proteins uncovered in basal fungi (*Figure 5B*, left). The first solved SBF/MBF crystal structure, Mbp1 from *S. cerevisiae* in the absence of DNA, originally suggested that Mbp1 recognizes its MCB (Mlu I cell cycle box, ACGCGT) binding site via a recognition helix (*Taylor et al., 1997*; *Xu et al., 1997*). However, a recent crystal structure of PCG2, an SBF/MBF homolog in the rice blast fungus *Magnaporthe oryzae*, in complex with its MCB binding site does not support this proposed mode of

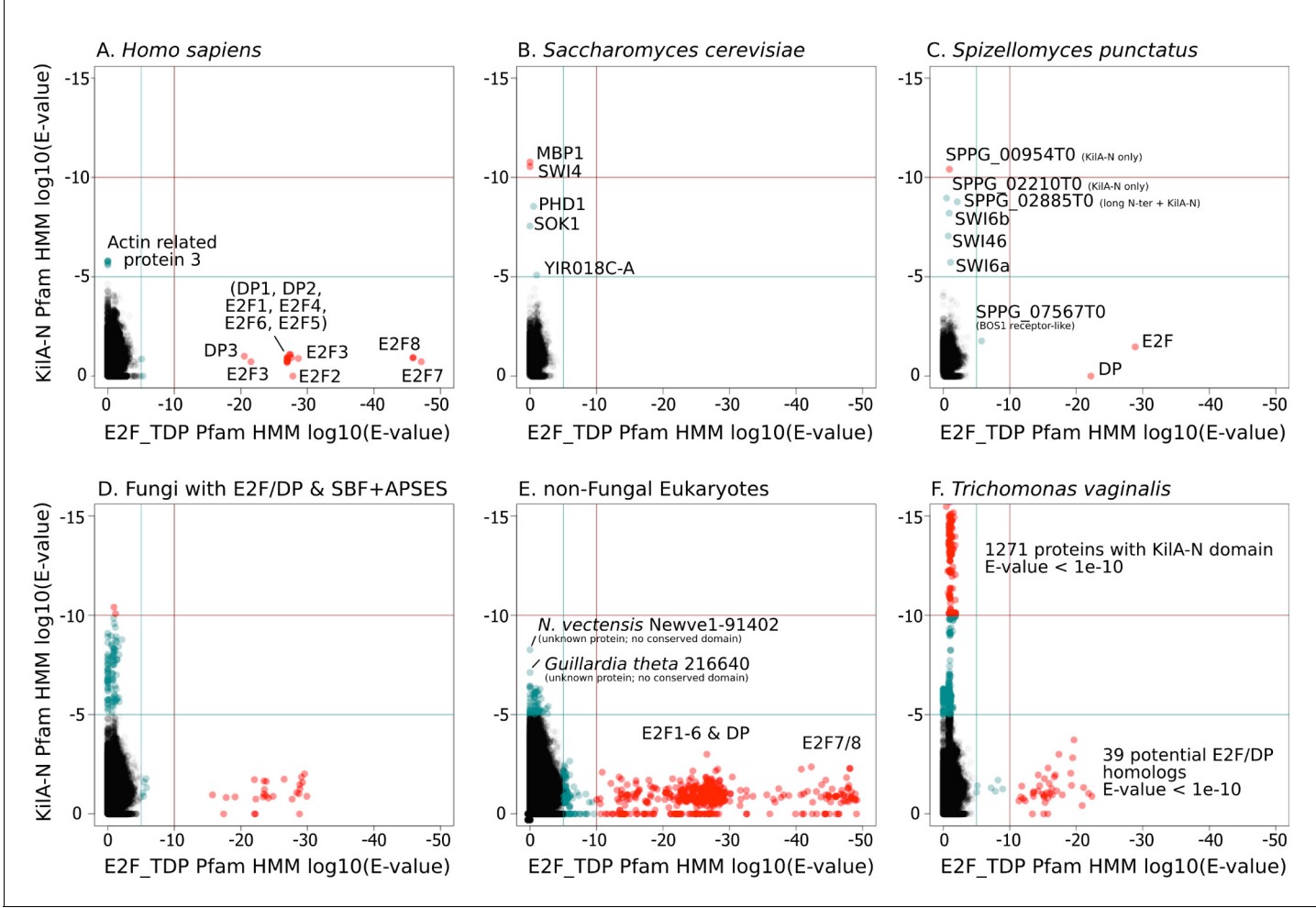

**Figure 4.** SBF and E2F HMM models detect different sequences. We used the Pfam HMMER model of the E2F/DP DNA-binding domain (E2F_TDP. hmm) and SBF DNA-binding domain (KilA-N.hmm). Every protein in the query genome (listed at top) was scored using hmmsearch with E2F/DP HMM (x-axis) and KilA-N HMM (y-axis). All scores below 1E-5 (i.e., marginally significant) are blue and those below 1E-10 (i.e. highly significant) are red. All hits with E-values between 1E-5 and 1E-10 were further validated (or rejected) using an iterative search algorithm (Jackhmmer) against the annotated SwissProt database using the HMMER web server (*Finn et al., 2011*). We then inspected these sequences manually for key conserved KilA-N residues. (A) *Homo sapiens* only has E2F/DP, (B) *Saccharomyces cerevisiae* only has KilA-N (i.e. SBF, MBF, APSES). (C) *Spizellomyces punctatus* and (D) other basal fungi have both E2F/DP and KilA-N. (E) All the non-fungal eukaryote genomes that we surveyed only have E2F/DP. (F) *Trichomonas vaginalis* is one of the few eukaryotes outside of fungi that has both E2F/DP and KilA-N. The E2F/DP HMM and KilA-N HMM always have orthogonal hits (i.e. no protein in our dataset significantly hits both HMMs).

DNA binding (*Liu et al., 2015*). In striking contrast to many wHTH structures, in which the recognition helix is the mediator of DNA binding specificity, the wing of PCG2 binds to the minor groove to recognize the MCB binding site. The two glutamines in the wing (Q82, Q89) are the key elements that recognize the core MCB binding motif CGCG (*Figure 5A*, blue structure). Family-specific conservation in the DNA-binding domain is observed for all members of the SBF family, including basal fungal sequences (*Figure 5B*, right). In summary, the incongruences in sequence, structure, and mode of DNA-interaction between E2F/DP and SBF/MBF families strongly suggest that SBF is not derived from E2F.

## Viral origin and evolution of the fungal SBF and APSES family

Since SBF is unlikely to be orthologous to the E2F family of transcription factors, we considered the straightforward alternative. Previous work has shown that the DNA-binding domain of the APSES and SBF proteins is homologous to a viral KilA-N domain (*Iyer et al., 2002*). KilA-N is a member of a

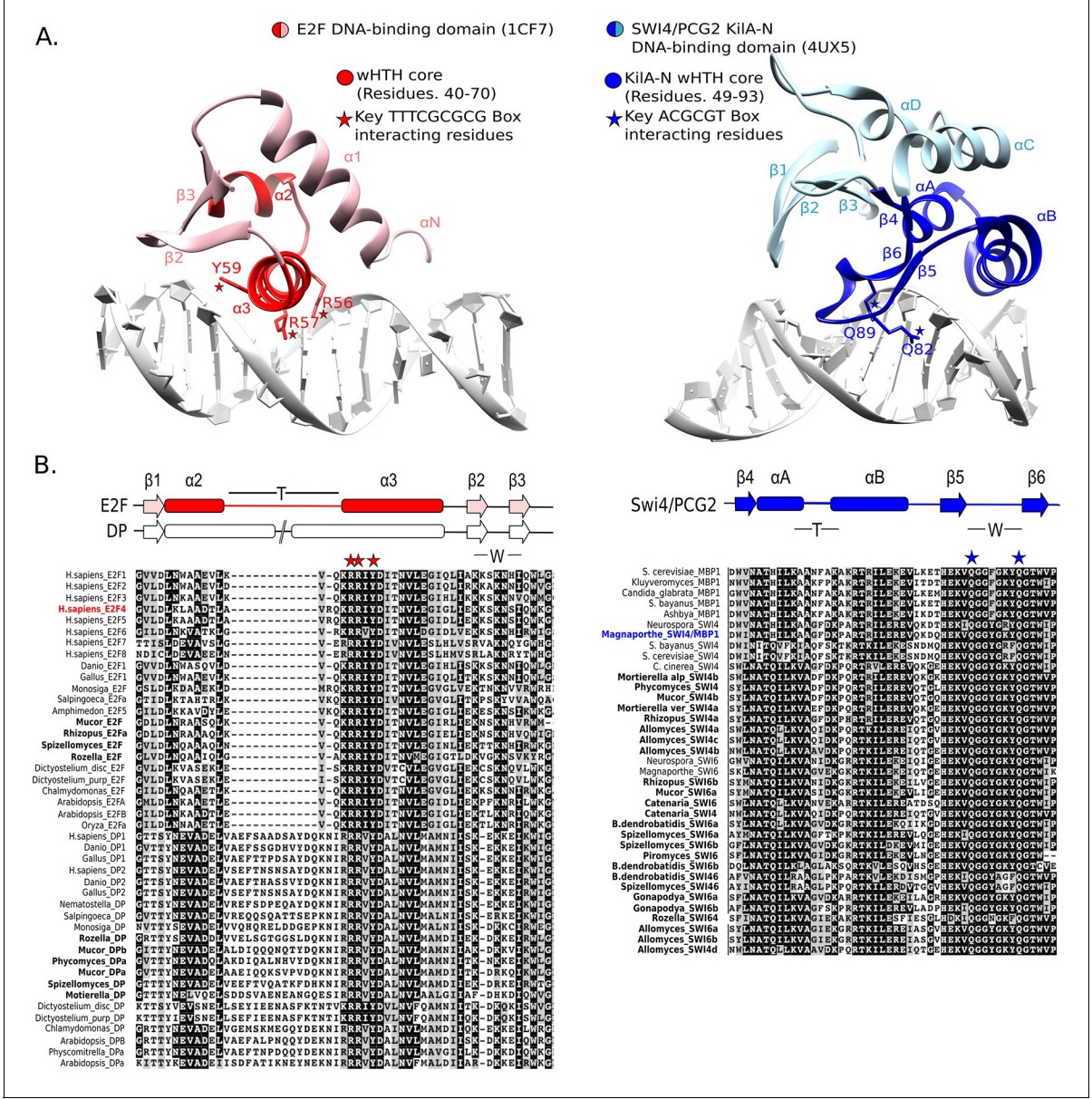

**Figure 5.** E2F and SBF show incongruences in sequence, structure, and mode of DNA binding. (**A**) Although both proteins share a winged helix-turn-helix (wHTH) domain, the E2F/DP and SBF/MBF superfamilies do not exhibit significant sequence identity or structural similarity to suggest a common recent evolutionary origin according to CATH or SCOP databases. Furthermore, each wHTH has a different mechanism of interaction with DNA: the arginine and tyrosine side-chains of recognition helix-3 of E2F (E2F4 from *Homo sapiens* [*Zheng et al., 1999*]) interact with specific CG nucleotides, where as the glutamine side-chains of the 'wing' of SBF/MBF (PCG2 from *Magnaporthe oryzae* [*Liu et al., 2015*]) interact with specific CG nucleotides. (**B**) Sequence alignment of the DNA binding domain of representative eukaryotic E2F/DP (left) and fungal SBF/MBF (right). The corresponding secondary structure is above the sequence alignment. Evolutionary conserved residues of sequence aligned DNA binding domains are highlighted in black. Bold sequence names correspond to E2F/DP and SBF/MBF sequences from basal fungi. Colored sequence names correspond to sequences of the structures shown in panel A. PDB IDs for the structures used are shown in parentheses. W = wing; T= turn.

core set of 'viral hallmark genes' found across diverse DNA viruses that infect eubacteria, archaea, and eukaryotes (*Koonin et al., 2006*). Outside the fungal SBF/APSES sub-family, little is known about the KilA-N domain structure, its DNA-binding recognition sequence, and function (*Brick et al., 1998*). The wide distribution of DNA viruses and KilA-N across the three domains of life suggests that the fungal ancestor likely acquired SBF via horizontal gene transfer rather than the other way around.

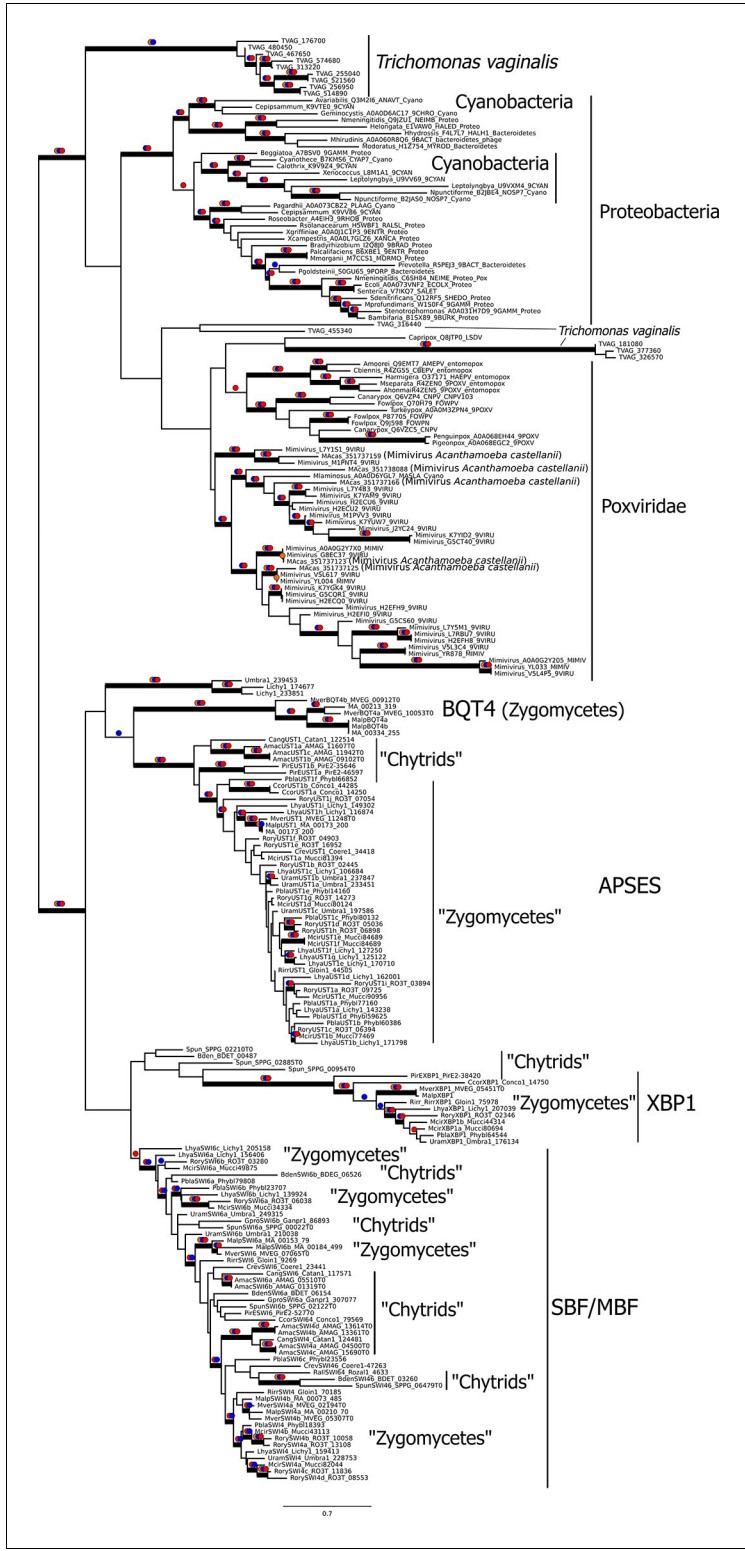

**Figure 6.** Viral origin of yeast cell cycle transcription factor SBF. Maximum likelihood unrooted phylogenetic tree depicting relationships of fungal SBF-family proteins, KilA-N domains in prokaryotic and eukaryotic DNA viruses. The original dataset was manually pruned to remove long-branches and problematic lineages. Our reduced KilA-N dataset has a total of 219 sequences (*Figure 6—source data 1*), 130 positions. Confidence at nodes was assessed with multiple support metrics using different phylogenetic programs under LG model of evolution. Colored dots in branches indicate corresponding branch supports (red dots: PhyML aBayes ≥0.9; blue dots:

*Figure 6 continued on next page*

*Figure 6 continued*

PhyML SH-aLRT ≥0.80; orange: RAxML RBS ≥70%). Thick branches indicate significant support by at least two metrics, one parametric and one non-parametric; scale bar in substitutions per site; see Materials and methods.
The following source data is available for figure 6:

**Source data 1.** Reduced set of KilA-N domains for phylogenetic analysis.

To broaden the scope of analysis beyond the eukaryotic genomes that we studied, we carefully surveyed all KilA-N domains present in the Pfam database. The majority of known KilA-N domains (weighted by species, not the number of sequences) are found in prokaryotes (85%) with a smaller fraction (10%) found in eukaryotes and a smaller fraction found in DNA viruses (5%). The KilA-N domains in prokaryotes appear to be either integrated by or derived from prokaryotic DNA viruses (i.e. bacteriophage), and thus, we will treat them as such. Within the eukaryotes, all known KilA-N domains are found in fungal genomes with three notable exceptions.

The first exception is *Trichomonas vaginalis*, a parasitic excavate with 1000+ KilA-N domains (*Figure 4F*). The *T. vaginalis* KilA-N domains have top blast hits to prokaryotic and eukaryotic DNA viruses, e.g. Mimivirus, a large double-stranded DNA virus of the Nucleo-Cytoplasmic Large DNA Viruses (*Yutin et al., 2009*). Mimiviruses are giant viruses known to infect simple eukaryotic hosts, such as *Acanthamoeba* and possibly other eukaryotes (*Abrahão et al., 2014*; *Raoult and Forterre, 2008*). The second and third exceptions are found in two insects, *Acyrthosiphon pisum* ('pea aphid') and *Rhodius prolixus* ('triatomid bug'). The one KilA-N domain in *A. pisum* genome has a top blast hit to eukaryotic DNA viruses (e.g. Invertebrate Iridescent Virus 6). The three KilA-N domains in *R. prolixus* have top blast hits to prokaryotic DNA viruses (e.g. Enterobacteria phage P1). The diverse and sparse distribution of KilA-N domains throughout the eukaryotic genomes is consistent with their horizontal gene transfer into hosts from eukaryotic DNA viruses and/or via engulfed bacteria that were infected with prokaryotic DNA viruses. In fact, the horizontal transfer of genes between Mimivirus and their eukaryotic host, or the prokaryotic parasites within the host, has been shown to be a more frequent event that previously thought (*Moreira and Brochier-Armanet, 2008*)

To gain further insight into the possible evolutionary origins of the SBF subfamily via horizontal gene transfer, we aligned diverse KilA-N sequences from the Uniprot and PFAM database to the KilA-N domain of our most basal fungal SBF+APSES sequences (Zoosporic fungi ('Chytrids') and 'Zygomycetes') and built a phylogenetic tree (*Figure 6*). There are three major phylogenetic lineages of KilA-N domains: those found in eukaryotic viruses, prokaryotic viruses, and the fungal SBF+APSES family. Our results show that the fungal SBF+APSES family is monophyletic and is strongly supported by multiple phylogenetic support metrics. This suggests a single HGT event as the most likely scenario that established the SBF+APSES family in a fungal ancestor. However, our current phylogeny is unable to distinguish whether the SBF family arrived in a fungal ancestor through a eukaryotic virus or a phage-infected bacterium. Structural and functional characterization of existing viral KilA-N domains could help distinguish between these two hypotheses.

## SBF ancestor could regulate E2F-target genes

Of all the members of the SBF+APSES family, the most likely candidate to be a 'founding' TF is SBF, as it is the only member present in all fungi (*Figure 3—figure supplement 1*). In budding yeast and other fungi, SBF functions in G1/S cell cycle regulation and binds a consensus site CGCGAA (*Gordân et al., 2011*), which overlaps with the consensus site GCGSSAAA for the E2F family (*Rabinovich et al., 2008*); see *Figure 7A*. The APSES regulators, Xbp1, and MBF in budding yeast bind TGCA, TCGA, ACGCGT motifs, respectively. A viral origin of the SBF+APSES family—with the founding member involved in cell cycle control—suggests the hypothesis that perhaps the founder TF functioned like a DNA tumor virus protein and hijacked cell cycle control to promote proliferation.

For the viral TF (SBF) to hijack cell cycle control in the fungal ancestor, it must have been able to both bind E2F regulatory regions and then activate the expression of genes under E2F in a cell cycle-regulated fashion. The overlap between the conserved E2F and SBF consensus sites suggests that ancestral SBF could bind E2F regulatory regions (*Figure 7A*). However, a single base pair

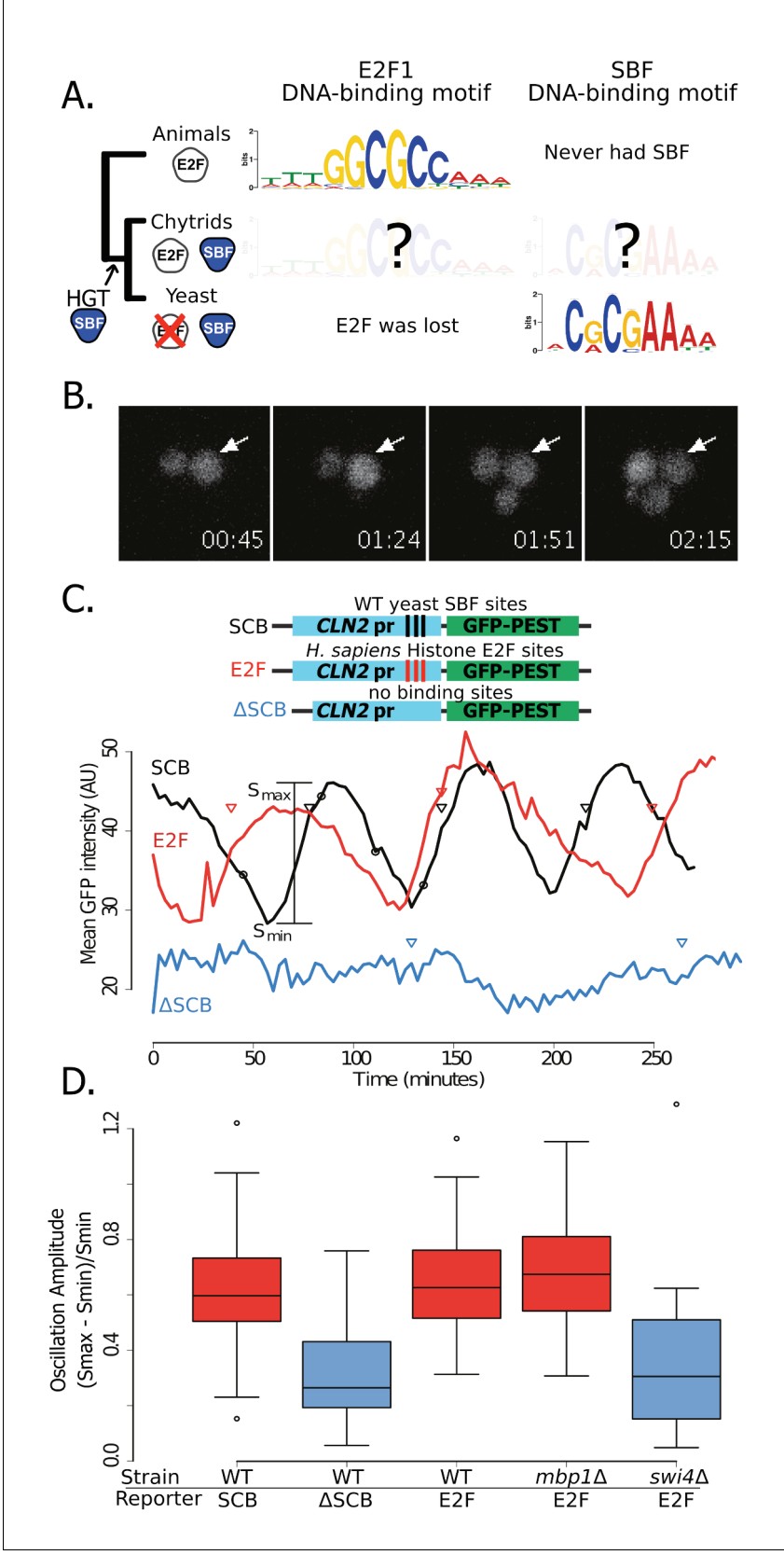

**Figure 7.** Yeast cell cycle transcription factor SBF can regulate cell cycle-dependent transcription via E2F binding sites in vivo. (**A**) Phylogenetic tree of animals, chytrids, yeast labelled with E2F, SBF or both transcription factors

*Figure 7 continued on next page*

*Figure 7 continued*

(TF) if present in their genomes. The known DNA-binding motifs of animal E2F (E2F1) and yeast SBF (Swi4) were taken from the JASPAR database, where as Chytrid E2F and SBF motifs are unknown. (**B**) Fluorescence images of cells expressing a destabilized GFP from the SBF-regulated *CLN2* promoter. (**C**) Oscillation of a transcriptional reporter in budding yeast. Characteristic time series of GFP expression from a *CLN2* promoter (SCB), a *CLN2* promoter where the SBF binding sites were deleted (ΔSCB), or a *CLN2* promoter where the SBF binding sites were replaced with E2F binding sites from the human gene cluster promoters (E2F). Oscillation amplitudes were quantified by scaling the mean fluorescence intensity difference from peak to trough divided by the trough intensity ($S_{max} - S_{min}$)/$S_{min}$. Circles denote time points corresponding to (**B**). Triangles denote budding events. (**D**) Distribution of oscillation amplitudes for different genotypes and GFP reporters. *swi4Δ* and *mbp1Δ* strains have deletions of the SBF and MBF DNA-binding domain subunits respectively. t-test comparisons within and across red and blue categories yield p-values >0.3 or <0.01 respectively. Boxes contain 25[th], median and 75[th] percentiles, while whiskers extend to 1.5 times this interquartile range.

substitution in the SBF motif can reduce gene expression by up to ∼95% (*Andrews and Moore, 1992*) and flanking regions outside the core are often important for binding affinity and gene expression (*Nutiu et al., 2011*). To our knowledge, no one has directly measured the extent to which animal E2F and yeast SBF bind similar sites either in vivo or in vitro. To first test whether yeast SBF can bind a canonical E2F binding site, we inserted consensus E2F binding sites in the budding yeast genome. The hijacking hypothesis would be supported in vivo if E2F binding sites could generate SBF-dependent cell cycle regulated gene expression. We used the well-studied *CLN2* promoter, which has three binding sites for SBF (SCB, Swi4,6-dependent cell cycle box) in a nucleosome-depleted region (*Figure 7B-C*). Removal of these SCB sites is known to eliminate cell cycle-dependent gene expression (*Bai et al., 2010*). We replaced the complete SBF sites (TCACGAAA) of *CLN2* (*Koch et al., 1996*) with a known E2F binding site (GCGCGAAA) from the promoters of the histone gene cluster in mammals (*Rabinovich et al., 2008*). We observed significant oscillations in GFP expression, which were coordinated with the cell cycle. Importantly, the amplitude of these oscillations was dependent on the budding yeast SBF (Swi4), but not MBF (Mbp1), and disappeared when the 3 binding sites were removed (*Figure 7C–D*). This experiment demonstrates that budding yeast SBF can bind E2F-like sites, despite the fact that Dikarya lost ancestral E2F hundreds of millions of years ago.

There are, of course, other possible E2F DNA binding sites that we could have used in our experiment; we picked this one because it is a well-characterized E2F binding site. To further explore the overlap in sequence specificity, we analyzed data from high-throughput protein-binding microarray (PBM) assays (*Afek et al., 2014*; *Badis et al., 2008*) of human E2F (E2F1) and budding yeast SBF (Swi4). PBM assays measure, in a single experiment, the binding of recombinant proteins to tens of thousands of synthetic DNA sequences, guaranteed to cover all possible 10-bp DNA sequences in a maximally compact representation (each 10-mer occurs once and only once). We used these PBM data to generate DNA motifs for E2F and SBF (*Berger et al., 2006*), and to compute, for each possible 8-bp DNA sequence, an enrichment score (or E-score) that reflects the specificity of the protein for that 8-mer. E-scores vary between -0.5 and +0.5, with larger values corresponding to higher affinity binding sites (*Berger et al., 2006*). As shown in *Figure 8*, E2F1 and Swi4 can bind a set of common motifs. For example, the E2F binding site variant that we tested in budding yeast (GCGCGAAA, highlighted in red), is one of the sites commonly bound in vitro by E2F and SBF.

Most notably, the in vitro PBM data show that there are specific motifs that can be bound only by E2F or only by SBF. To identify the key nucleotide differences between E2F-only and SBF-only binding, we created motifs of E2F-only and SBF-only sites. The consensus E2F-only (NN<u>S</u>GCGSN) and SBF-only (NN<u>CR</u>CGNN) motifs indicate that differential specificity between E2F1 and SBF is mediated by the nucleotides in the 3rd and 4th positions (underlined) before the invariant CG at the 5th and 6th positions. E2F has a strict preference for G in the 4th position, where as SBF has a strict preference for C in the 3rd position (*Figure 8*).

We then scanned the promoters of known E2F target genes from the human genome (*CCNE1, E2F1, EZH2*) with our empirically-defined DNA binding sites from PBM assays (*Afek et al., 2014*; *Badis et al., 2008*) to predict putative E2F-only, SBF-only, and common sites (*Figure 8—figure supplement 1*). As expected, there are many predicted E2F-only and common (E2F & SBF) sites that

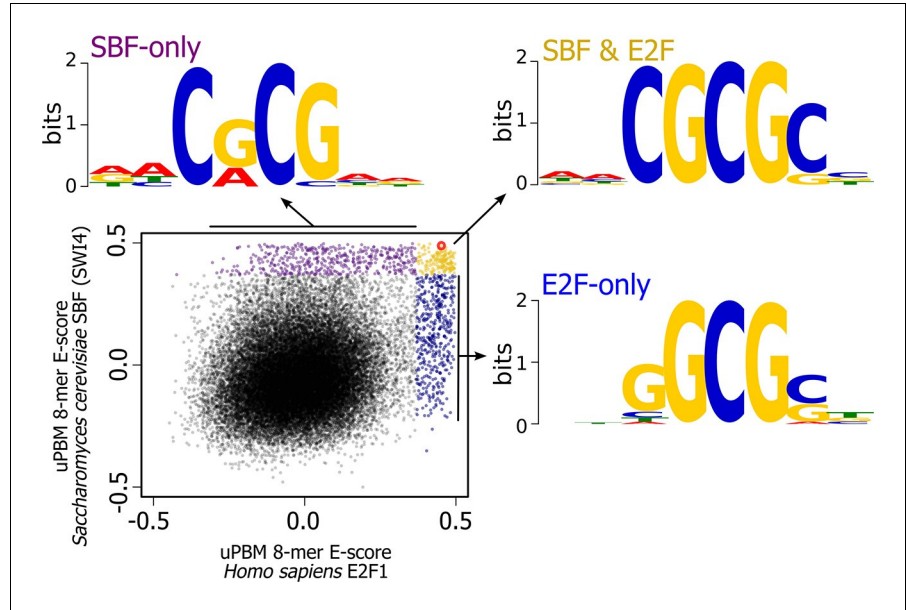

**Figure 8.** High-throughput DNA binding data for yeast SBF and human E2F shows that SBF and E2F can bind shared and distinct DNA-binding sites. Plot of in vitro protein binding microarray 8-mer E-scores for *Homo sapiens* E2F1 (*Afek et al., 2014*) versus *S. cerevisiae* SBF protein Swi4 (*Badis et al., 2008*). All 8-mer motifs colored (E-score > 0.37) are considered significant targets with a false positive discovery rate of 0.001 (*Badis et al., 2009*). Yellow are common 8-mer motifs bound by both E2F1 and SBF, blue are E2F-only motifs, and purple are SBF-only motifs. The E2F motif from histone cluster promoters used in *Figure 7* is circled in red.

The following figure supplements are available for figure 8:

**Figure supplement 1.** Bioinformatic scan of E2F-regulated human promoters suggests possible regulation by SBF.

**Figure supplement 2.** Many E2F-regulated genes in humans could be bound by SBF.

could be bound by E2F in these known target genes. However, we could also find many common and SBF-only binding sites to which SBF could bind. We then extended our analysis to 290 known E2F target genes in the human genome to test the generality of SBF cross-binding to E2F sites (*Eser et al., 2011*). Most E2F target promoters could be bound by SBF (*Figure 8—figure supplement 2*). Taken together, this set of experiments lends support to the hijacking hypothesis, where an ancestral SBF may have taken control of several E2F-regulated genes.

## Discussion

Cell division is an essential process that has been occurring in an uninterrupted chain for billions of years. Thus, one expects strong conservation in the regulatory network controlling the eukaryotic cell division cycle. Consistent with this idea, cell cycle network structure is highly similar in budding yeast and animal cells. However, many components performing similar functions, such as the SBF and E2F transcription factors, lack sequence identity, suggesting a significant degree of evolution or independent origin. To identify axes of conservation and evolution in eukaryotic cell cycle regulation, we examined a large number of genome sequences in Archaeplastida, Amoebozoa, SAR, Haptophyta, Cryptophyta, Excavata, Metazoa and Fungi. Across eukaryotes, we found a large number of proteins homologous to metazoan rather than fungal G1/S regulators. Our analysis indicates that the last eukaryotic common ancestor likely had complex cell cycle regulation based on Cdk1, Cyclins D, E, A and B, E2F, pRb and APC family proteins.

In contrast, SBF was not present in the last common eukaryotic ancestor, and abruptly emerged, with its regulator Whi5, in fungi likely due to the co-option of a viral KilA-N protein at the base of

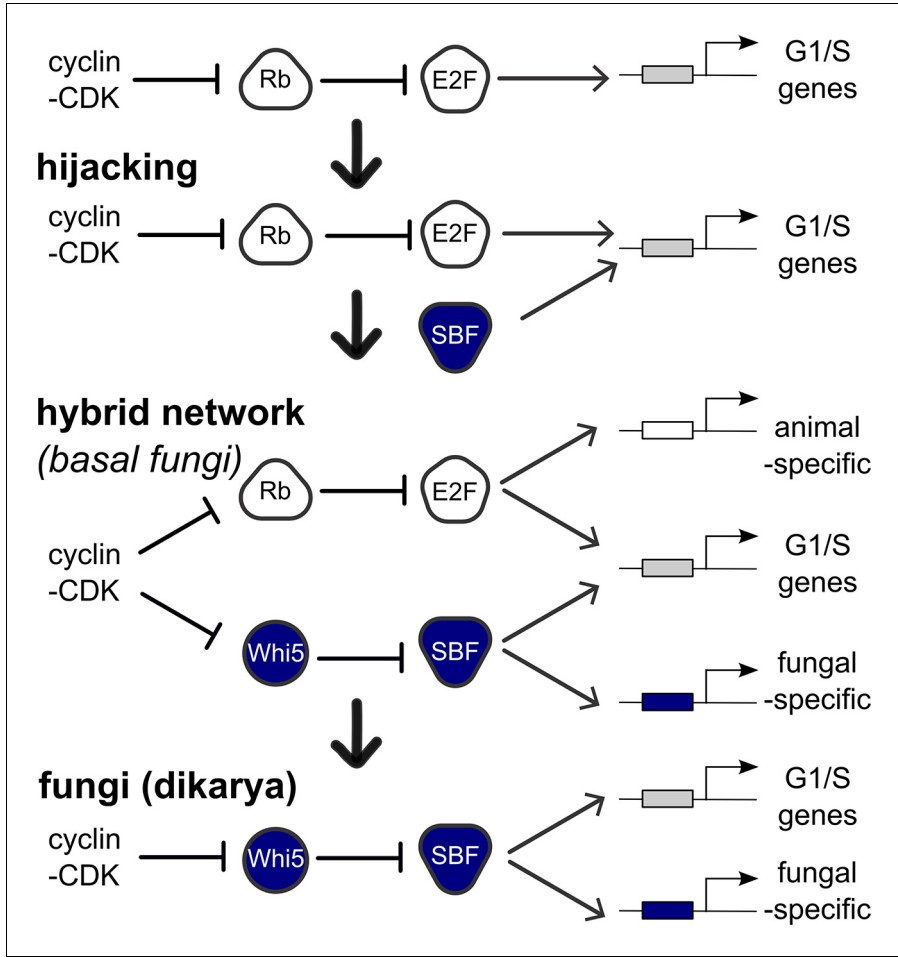

**Figure 9.** Punctuated evolution of a conserved regulatory network. Evolution can replace components in an essential pathway by proceeding through a hybrid intermediate. Once established, the hybrid network can evolve dramatically and lose previously essential regulators, while sometimes retaining the original network topology. We hypothesize that SBF may have hijacked the cell cycle of a fungal ancestor by binding cis-regulatory DNA sites of E2F and activating expression of G1/S genes, thus promoting cell cycle entry. Cell cycle hijacking in a fungal ancestor was followed by evolution of Whi5 to inhibit SBF and Whi5 was subsequently entrained to upstream cell cycle control through phospho-regulation by old or new cyclin-CDKs to create a hybrid network with parallel pathways. The hybrid network likely provided redundant control of the G1/S regulatory network, which could explain the eventual loss of E2F and its replacement by the SBF pathway in more derived fungi (Dikarya). Interestingly, zoosporic fungi such as Chytrids have hybrid networks and are transitional species because they exhibit animal-like features of the opisthokont ancestor (centrioles, flagella) and fungal-like features (cell wall, hyphal growth). We hypothesize that E2F and SBF also bind and regulate a subset of animal-specific and fungal-specific G1/S genes, which could help explain the preservation of the hybrid network in Chytrids. Ancestral SBF expanded to create an entire family of transcription factors (APSES) that regulate fungal-specific traits such as sporulation, differentiation, morphogenesis, and virulence.

the fungal lineage. The origin of Whi5 is unclear because we found no homologs outside of fungi. Whi5 is a mostly unstructured protein, which, like pRb, recruits transcriptional inhibitor proteins to specific sites on DNA via transcription factor binding (*Huang et al., 2009*; *Travesa et al., 2013*; *Wang et al., 2009*). The relatively simple structure of Whi5 suggests that it may have been subsequently co-opted as a phosphopeptide to entrain SBF activity to cell cycle regulated changes in Cdk activity (*Figure 9*).

The replacement of E2F-Rb with SBF-Whi5 at the core of the cell cycle along the fungal lineage raises the question as to how such a drastic change to fundamental regulatory network could evolve. One answer can be found in the evolution of transcription factors. When the function of an essential

transcription factor does change, it often leaves behind a core part of its regulon for another factor (*Brown et al., 2009*; *Gasch et al., 2004*; *Lavoie et al., 2009*). This process of handing off transcription factor function has been observed to proceed through an intermediate state, present in some extant genomes, in which both factors perform the function (*Tanay et al., 2005*). The logic of proceeding through an intermediate state has been well-documented for the regulation of genes expressed only in yeast of mating type **a** (asgs) (*Tsong et al., 2006*). In the ancestral yeast and many extant species, asgs expression is activated by a protein only present in **a** cells, while in other yeasts, expression is repressed by a protein only present in α cells and **a**/α diploids. The replacement of the ancestral positive regulation by negative regulation occurred via yeast that contained both systems illustrating how an essential function can evolve through a hybrid state (*Baker et al., 2012*).

Clearly, something similar happened during cell cycle evolution. It appears that the replacement of the E2F-pRb transcription regulatory complex with the SBF-Whi5 complex proceeded via a hybrid intermediate that preserved its function. In the hybrid intermediate, E2F-Rb and SBF-Whi5 may have evolved to be parallel pathways whose functions overlapped to such an extent that the previously essential E2F-Rb pathway could be lost in the transition to Dikarya. Interestingly, many basal fungi (e.g. Chytrids) have preserved rather than lost this hybrid intermediate, which suggests that each pathway may have specialized functions. Chytrids exhibit both animal (e.g. centrioles, flagella, amoeboid movement) and fungal features (e.g. cell wall, hyphal growth) whose synthesis needs to be coordinated with cell division. The preservation of the hybrid network in chytrids could then be explained if animal-like and fungal features are regulated by the E2F-Rb and SBF-Whi5 pathways respectively (*Figure 9*). Once Fungi lost many of the ancestral animal-like features during the emergence of the 'zygomycetes' or Dikarya, the ancestral E2F-pRb components could have evolved new functions or have been lost.

The origin of the hybrid network at the base of Fungi is abrupt and may have been initiated by the arrival of SBF via virus. Many tumor viruses activate cell proliferation. For example, the DNA tumor viruses Adenovirus and SV40 highjack cell proliferation in part by activating the expression of E2F-dependent genes by binding pRb to disrupt inhibition of E2F (*DeCaprio, 2009*). While the specific mechanisms may differ, when SBF entered the fungal ancestor cell it might have activated the transcription of E2F target genes. Rather than inhibiting the inhibitor of E2F, SBF may have directly competed for E2F binding sites with transcriptionally inactive E2F-Rb complexes (*Figure 9*). Consistent with this model, we have shown here that SBF can directly regulate gene expression in budding yeast via a consensus E2F binding site. Thus, the cooption of a viral protein generated a hybrid network to ultimately facilitate dramatic evolution of the core cell cycle network in fungi.

## Materials and methods

### Identification of potential protein family homologs

We used Profile-Hidden Markov Models (profile-HMMs) to detect homologs for each of the families studied, using the HMMER 3 package (*Eddy, 2011*). Profile-HMMs are sensitive tools for remote homology detection. Starting with a set of diverse yet reliable protein homologs is fundamental for detecting remote protein homology and avoiding 'model poisoning' (*Johnson et al., 2010*). To this end, we used reliable training-set homologs from the cell cycle model organisms *Arabidopsis thaliana*, *Homo sapiens*, *Schizosaccharomyces pombe*, and *Saccharomyces cerevisiae*, to build the profile-HMMs used to detect homologs. Our profile-HMM search used a stringent E-value threshold of 1E-10 to detect putative homologs in the 'best' filtered protein sets (where available) of our 100+ eukaryotic genomes (see *Supplementary file 1A* for genome details). All putative homologs recovered through a profile-HMM search were further validated (or rejected) using an iterative search algorithm (Jackhmmer) against the annotated SwissProt database using the HMMER web server (*Finn et al., 2011*).

Our profile-HMM for E2F/DP family only detects E2F or DP, where as our profile-HMM for SBF/MBF family only detects SBF/MBF (or APSES). The same protein was never identified by both profile-HMMs because the sequence profiles and the structure are non-homologous. In the case of basal fungi, which have both E2F/DP and SBF/MBF, all proteins classified as an E2F/DP had clear homology to E2F or DP (see alignment in *Figure 5B*) and all proteins that we classified as SBF/MBF had clear homology to SBF/MBF (see alignment in *Figure 5B*).

## Phylogenetic-based classification of protein homologs in sub-families

A phylogenetic analysis and classification was built in four stages. In the first stage, we used MAFFT-L-INS-i (-maxiterate 1000) to align the sequences of eukaryotic protein family members (*Katoh and Standley, 2013*). We then used probabilistic alignment masking using ZORRO (*Wu et al., 2012*) to create different datasets with varying score thresholds. Next, we used ProtTest 3 to determine the empirical amino-acid evolutionary model that best fit each of our protein datasets using several criteria: Akaike Information Criterion, corrected Akaike Information Criterion, Bayesian Information Criterion and Decision Theory (*Darriba et al., 2011*). Last, for each dataset and its best-fitting model, we ran different phylogenetic programs that use maximum-likelihood methods with different algorithmic approximations (RAxML and PhyML) and Bayesian inference methods (PhyloBayes-MPI) to reconstruct the phylogenetic relationships between proteins.

For RAxML analyses, the best likelihood tree was obtained from five independent maximum likelihood runs started from randomized parsimony trees using the empirical evolutionary model provided by ProtTest. We assessed branch support via rapid bootstrapping (RBS) with 100 pseudo-replicates. PhyML 3.0 phylogenetic trees were obtained from five independent randomized starting neighbor-joining trees (RAND) using the best topology from both NNI and SPR moves. Non-parametric Shimodaira-Hasegawa-like approximate likelihood ratio tests (SH-aLRTs) and parametric *à la Bayes* aLRTs (aBayes) were calculated to determine branch support from two independent PhyML 3.0 runs. For Bayesian inference we used PhyloBayes (rather than the more frequently used MrBayes) because it allows for site-specific amino-acid substitution frequencies, which better models the level of heterogeneity seen in real protein data (*Lartillot and Philippe, 2004*; *Lartillot et al., 2009*). We performed Phylobayes analyses by running three independent chains under CAT and the exchange rate provided by ProtTest 3 (e.g. CAT-LG), four discrete gamma categories, and with sampling every 10 cycles. Proper mixing was initially confirmed with Tracer v1.6 (2014). The first 1000 samples were discarded as burn-in, and convergence was assessed using bipartition frequencies and summary statistics provided by bpcomp and tracecomp from Phylobayes. These were visually inspected with an R version of AWTY (https://github.com/danlwarren/RWTY) (*Nylander et al., 2008*). The best phylogenies are shown in *Figure 2—figure supplement 2–6* and *Figure 3—figure supplement 2–4*, and were used to tentatively classify sequences into sub-families and create *Figure 2—figure supplement 1* and *Figure 3—figure supplement 1*.

We note that the confidence of each node in the phylogenetic trees was assessed using multiple, but complementary support metrics: (1) posterior probability for the Bayesian inference, (2) rapid bootstrap support (*Stamatakis, 2006*; *Stamatakis et al., 2008*) for RAxML, and (3) non-parametric Shimodaira-Hasegawa-like approximate likelihood ratio tests (SH-aLRTs) and parametric *à la Bayes* aLRTs (aBayes) for PhyML. These different support metrics complement each other in their advantages and drawbacks. SH-aLRT is conservative enough to avoid high false positive rates but performs better compared to bootstrapping (*Guindon et al., 2010*; *Simmons and Norton, 2014*). aBayes is powerful compared to non-parametric tests, but has a tendency to increase false-positive rates under serious model violations, something that can be balanced with SH-aLRTs (*Anisimova and Gascuel, 2006*; *Anisimova et al., 2011*).

## Strain construction

Our *CLN2pr-GFP-CLN2PEST* constructs were all derived from pLB02-0mer (described in *Bai et al., 2010* and obtained from Lucy Bai). To create pLB02-CLN2, a synthetic DNA fragment (IDT, Coralville, IA) encompassing a region of the *CLN2* promoter from 1,130 bp to 481 bp upstream of the *CLN2* ORF was digested with BamHI and SphI and ligated into pLB02-0mer digested with the same enzymes. To create pLB02-E2F, which contains E2F binding sites, the same procedure was applied to a version of the promoter fragment in which the SCBs at 606bp, 581bp, and 538bp upstream of the ORF were replaced with the E2F binding site consensus sequence GCGCGAAA (*Rabinovich et al., 2008*). All these plasmids were linearized at the BbsI restriction site in the *CLN2* promoter and transformed. Both *swi4Δ* and *mbp1Δ* strains containing pLB02-0mer, pLB02-Cln2, pLB02-E2F fluorescent expression reporters were produced by mating lab stocks using standard methods. JE103 was a kind gift from Dr. Jennifer Ewald. Plasmids and strains are listed in *Supplementary file 1B and 1C*, respectively.

## Imaging and analysis

Imaging proceeded essentially as described in *Bean et al., 2006* . Briefly, early log-phase cells were pre-grown in SCD and gently sonicated and spotted onto a SCD agarose pad (at 1.5%), which was inverted onto a coverslip. This was incubated on a heated stage on a Zeiss Observer Z.1 while automated imaging occurred (3 min intervals, 100–300 ms fluorescence exposures). Single-cell time-lapse fluorescence intensity measurements were obtained using software described in *Doncic and Skotheim (2013)*, *Doncic et al. (2011)*, and oscillation amplitudes were obtained manually from the resulting traces. The single-cell fluorescence intensity traces used mean cellular intensity with the median intensity of the entire field of view subtracted, to control for any fluctuations in fluorescent background. The resulting measurements were analyzed in R.

## Bioinformatic analysis of protein binding microarrays and human promoters

Universal PBM data for Swi4 was downloaded from the cis-BP database (*Weirauch et al., 2014*). We used the PBM 8-mer E-scores reported in cis-BP for the data set M0093_1.02:Badis08: SWI4_4482.1_ArrayB. Universal PBM data for E2F1 (*Afek et al., 2014*) is available in the GEO database (accession number GSE61854, probes UnivV9_*). E2F1 8-mer E-scores were computed using the Universal Protein Binding Microarray (PBM) Analysis Suite (*Berger and Bulyk, 2009*). An E-score cutoff of 0.37 was used to call SBF and E2F binding sites. This cutoff corresponds to a false positive discovery rate of 0.001 (*Badis et al., 2009*). To generate DNA motifs for E2F-only, SBF-only, and common sites, we used the Priority software (*Gordân et al., 2010*) with a uniform prior to align the 8-mers with E-score > 0.37 for E2F only, SBF only, or both E2F and SBF, respectively (*Figure 8*). Promoter sequences (1000 bp upstream of transcription start) for known E2F targets (*Eser et al., 2011*) were retrieved from *Homo sapiens* genome as provided by UCSC (hg38) (BSgenome.Hsapiens. UCSC.hg38) and the annotation package org.Hs.eg.db from Bioconductor (v3.2) (*Gentleman et al., 2004*; *Huber et al., 2015*). Only regions for which at least two consecutive sliding windows of 8-mers (1 nucleotide step; 7 overlapping nucleotides) were high scoring (E-score $\geq$ 0.37) were called as potential SBF or E2F binding regions. Overlapping or 'common' binding regions between E2F and SBF were defined as regions that, regardless of their difference in length (nested or partially overlapping), overlapped in at least one full 8-mer (*Figure 8—figure supplement 1*).

## Acknowledgements

We thank F Cross and A Robinson-Mosher for extensive discussions, and J Heitman, D Lew, J Nevins, S Rubin for critical comments on the manuscript. We thank J Stajich for providing access to the Bioinformatics cluster at the Institute for Integrative Genome Biology at UC Riverside supported by the College of Natural and Agricultural Sciences, the National Science Foundation, and Alfred P Sloan Foundation.

## Additional information

### Funding

| Funder | Grant reference number | Author |
| --- | --- | --- |
| National Institutes of Health | 1DP2OD008654-01 | Edgar M Medina<br>Nicolas E Buchler |
| Burroughs Wellcome Fund | | Edgar M Medina<br>Jonathan J Turner<br>Jan M Skotheim<br>Nicolas E Buchler |
| Defense Advanced Research Projects Agency | BAA-11-66 | Edgar M Medina<br>Nicolas E Buchler |
| National Institutes of Health | GM092925 | Jonathan J Turner<br>Jan M Skotheim |
| Alfred P. Sloan Foundation | | Raluca Gordân |

National Science Foundation     MCB-14-12045,          Raluca Gordân

The funders had no role in study design, data collection and interpretation, or the decision to submit the work for publication.

## Author contributions

EMM, JJT, JMS, NEB, Conception and design, Acquisition of data, Analysis and interpretation of data, Drafting or revising the article; RG, Analysis and interpretation of data, Drafting or revising the article

## Author ORCIDs

Edgar M Medina, http://orcid.org/0000-0002-5518-5933
Jonathan J Turner, http://orcid.org/0000-0001-7311-0313
Raluca Gordân, http://orcid.org/0000-0002-6404-6556
Jan M Skotheim, http://orcid.org/0000-0001-8420-6820
Nicolas E Buchler, http://orcid.org/0000-0003-3940-3432

# Additional files

## Supplementary files

• Supplementary file 1. (A) List of eukaryotic genomes. We downloaded and analyzed the following annotated genomes using the 'best' filtered protein sets when available. We gratefully acknowledge the Broad Institute, the DOE Joint Genome Institute, Génolevures, PlantGDB, SaccharomycesGD, AshbyaGD, DictyBase, JCV Institute, Sanger Institute, TetrahymenaGD, PythiumGD, AmoebaDB, NannochloroposisGD, OrcAE, TriTryDB, GiardiaDB, TrichDB, CyanophoraDB, and CyanidioschizonDB for making their annotated genomes publicly available. We especially thank D. Armaleo, I. Grigoriev, T. Jeffries, J. Spatafora, S. Baker, J. Collier, and T. Mock for allowing us to use their unpublished data. (B) Plasmids. (C) Strains. All yeast strains were derived from W303 and constructed using standard methods.

## Major datasets

The following previously published datasets were used:

| Author(s) | Year | Dataset title | Dataset URL | Database, license, and accessibility information |
|---|---|---|---|---|
| Afek A, Schipper JL, Horton J, Gordân R, Lukatsky DB | 2014 | Protein−DNA binding in the absence of specific base-pair recognition | http://www.ncbi.nlm.nih.gov/geo/query/acc.cgi?acc=GSE61854 | Publicly available at NCBI Gene Expression Omnibus (accession no. GSE61854) |
| Weirauch MT, Yang A, Albu M, Cote AG, Montenegro-Montero A, Drewe P, Najafabadi HS, Lambert SA, Mann I, Cook K, Zheng H, Goity A, van Bakel H, Lozano JC, Galli M, Lewsey MG, Huang E, Mukherjee T, Chen X, Reece-Hoyes JS, Govindarajan S, Shaulsky G, Walhout AJ, Bouget FY, Ratsch G, Larrondo LF, Ecker JR, Hughes TR | 2014 | Determination and inference of eukaryotic transcription factor sequence specificity | http://cisbp.ccbr.utoronto.ca/ | Publicly available at the CIS-BP Database (M0093_1.02:Badis08:SWI4_4482.1_ArrayB) |

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
