## [Decision Letter]

Thank you for submitting your work entitled "Punctuated evolution and transitional hybrid network in an ancestral cell cycle of fungi" for peer review at *eLife*. Your submission has been evaluated by Diethard Tautz (Senior editor), and three reviewers, one of whom is a member of our Board of Reviewing Editors.

The reviewers have discussed the reviews with one another and the Reviewing editor has drafted this decision to help you prepare a revised submission.

Summary:

This paper addresses the evolution of cell cycle specific transcription factors and cyclins in fungi. All of us feel that the cyclin part was less interesting, more speculative, and poorly motivated. Your story about evolution of SBF-like transcription factors is the more interesting aspect. Unfortunately, the key discovery discussed in the paper, namely the origin of SBF from DNA viruses was reported long ago by Koonin and this aspect of the paper is therefore not entirely novel. More interesting is the finding that basal fungi appear to have both SBF and E2F/Rb like factors, confirming that E2F/Rb must have been replaced by SBF early on in the fungal lineage. In a way, one already knew that this must be case since the E2F system is present in plants as well as animals, implying that it must have been present in the common ancestor of fungi and animals, which are closer to each other than plants. We already knew therefore that E2F must therefore have been lost early in the fungal lineage. However, placing when this took place is certainly interesting and an important advance in our understanding.

All of us were unimpressed by the experiment that E2F site from a histone promoter in mammals can replace the SBF sites in the CLN2 promoter. Results are also consistent with the obvious alternative hypothesis that SBF and E2Fs are distant homologues and the binding specificity has simply been conserved during evolution. We don't see how any experiment along these lines could be said to support the hijacking hypothesis (to the exclusion of the obvious competing hypothesis). Furthermore, even if SBF and E2F are non-homologous, they have overlapping binding specificity. Since sequence specific transcription factors allow degeneracy (or wobble positions) and bind to short DNA sequence families (motifs), these experiments show only that the E2F sites that were chosen probably overlap the specificity of SBF enough that they can be regulated. Therefore, these experiments really add nothing to the hijacking model, except to confirm the observation that the two transcription factors have overlapping specificity.

Overall, the paper has one really key novel observation, namely the detection of the E2F/Rb system in basal fungi and this is certainly worth publishing. However, you need to show that there really are both classes of proteins in the basal fungi. You need to consider and rule out alternative explanations. For example, are you simply assigning some of the homologues in the basal fungi to be E2Fs and some to be SBF/MBFs because they have reached the limit of homology detections?

The problem is that this key finding has been wrapped up in a paper that merely repeats Koonin's findings and addresses the evolution of fungal cyclins in a manner that is not particularly revealing. We would recommend a far shorter paper focusing on the one key finding.

Other Major Comments:

Note some of these points may not be relevant when you shorten this article. Nonetheless, these comments may help if you decide to wrap up the remaining content for another journal:

Reviewer 1:

With regard to your models in Figure 7 and Figure 8, have you thought about alternative hypotheses? For example, CLN sets the stage for the acquisition of SBF. What evidences do you have to rule alternatives out?

Reviewer 3:

1) Figure 3 and Figure 4 are critical to the authors' argument, but the data are very poorly presented. The whole argument rests on the accurate detection of homologues for both types of regulators in the basal fungi. The authors need to take much more care to convince the reader that the observations are not simply due to errors in the homology detection (due for example to longer branches or denser sampling of species in the fungi).

For example, they need to show convincing evidence that the diversity of the SBF/MBFs falls outside of the diversity of the E2Fs, and that the viral proteins fall within the SBFs and outside of the E2Fs. Figure 6a shows a phylogeny of SBFs, but it doesn't include the E2Fs.

Even more important, they need to show that there really are both classes of proteins in the basal fungi. They need to consider and rule out alternative explanations. For example, are they simply assigning some of the homologues in the basal fungi to be E2Fs and some to be SBF/MBFs because they have reached the limit of homology detections?

2) The structural evidence could represent key supporting data to show that E2Fs and SBFs really are two different families. However, is not presented in a very convincing way.

"it is possible that fungal SBF is a duplicated and hyper-evolved E2F. However, the number and topology of secondary elements (i.e., α helices, β strands, and loops) that stabilize and pack around the core helix-turn-helix are different."

Does the number and topology of secondary structural elements always stay the same in homologues? If not, how does this support the hypothesis that they are non-homologous?

"Crystal structures of the DNA binding domains of E2F and SBF show that they are not structural homologues (Taylor et al., 1997; Xu et al., 1997) (see Figure 5)."

Here the authors just state that E2F and SBF are not structural homologues. What is it about the structures in Figure 5 that shows that they are not homologues?

3) Even if the structural evidence is sound, and the proteins are non-homologous, the authors specific argument rests on the identification of *both* proteins in the basal fungi. The authors should show and explain the evidence that the putative homologues in the basal fungi are likely to adopt one structure and not the other. The alignment in Figure 5 shows many irrelevant species, and the homologues from the basal fungi are not highlighted.

I think the authors need to consider their evolutionary model much more carefully and design experiments that actually test it.

[Editors' note: further revisions were requested prior to acceptance, as described below.]

Thank you for submitting your work entitled "Punctuated evolution and transitional hybrid network in an ancestral cell cycle of fungi" for consideration by *eLife*. Your article has been reviewed by two peer reviewers, and the evaluation has been overseen by a Reviewing Editor and Diethard Tautz as the Senior Editor.

The reviewers have discussed the reviews with one another and the Reviewing Editor has drafted this decision to help you prepare a revised submission.

The reviewers see improvements in your revision but are still far from being satisfied. Ordinarily, this level of criticism on a revised submission would be considered unacceptable, however, we have decided to provide you with an opportunity to substantially revise the manuscript one last time. If you have questions about our instructions, please write and ask for clarification. Of course, given the concerns we have expressed we would understand if you choose to submit to a different journal. Please advise as to your course of action as soon as possible.

I encourage you to shorten this further (as per multiple reviewers' request).

Reviewer #1:

The authors did a good job of clarifying the structural data and present a much more convincing case that the two proteins are non-homologous.

However, they did very little to address my other concerns.

Rather than doing a detailed, focused analysis to convince the reader of their hypothesis that the basal fungi contain both transcription factors, and that the SBFs are inherited from a single horizontal transfer event, they simply bolded the basal fungi species names in an alignment. They did not show any phylogenetic analysis that would rule out competing evolutionary scenarios (such as recent transfer of the E2Fs into the basal fungi) or technical issues (such as homology detection problems). In response to my comments about the latter, they simply stated that their HMMs don't make any classification errors:

"Our profile-HMM for E2F/DP family only detects E2F or DP, where as our profile-HMM for SBF/MBF family only detects SBF/MBF (or APSES)."

In fact, I agree with the authors that the HMMs are probably very accurate, but why not show data to support this claim? (E.g., the score distributions or phylogenetic trees?)

The authors don't seem to appreciate (or have chosen to ignore) the fact that remote homology detection (especially the claim that homologs are *not* present) is a statistical process and always has an element of uncertainty. (For example, there might be additional diverged SBF homologues in other eukaryotes that are below the E-value threshold, missed due to gene prediction errors, incomplete genome sequences, etc.) Since the detection of these homologs in the basal fungi and no SBFs in other eukaryotes is the key result of this paper, why not make a convincing case to support it (instead of the unnecessary Figure 2 and Figure 3)?

Similarly, they have not changed the presentation of the experiment demonstrating that the SBF can bind to an E2F binding site. In their response, they claim that this experiment is important because "a single base pair substitution in the SBF motif can reduce gene expression". However, they have still not showed *which* substitution that was. Therefore the argument is misleading: all substitutions in transcription factor binding sites are not equivalent, and this experiment would only be interesting if the substitution between the histone binding sites and the SBF consensus was one of those that was likely to reduce expression. If it is, why not show this and explain it clearly?

Furthermore, since E2Fs regulate many cell cycle target genes, it would be much more convincing if the authors simply surveyed the known E2F binding sites and estimated (bioinformatically) which of them SBF is likely to regulate. The observation that SBF can bind the histone binding site (on its own) really adds nothing.

I think the most important thing for the authors to add is a phylogenetic analysis showing that the data support the hypothesis of a single transfer event in the ancestral fungi.

As a secondary point, I guess if they really want to include the histone binding site experiment, they should include it in the context of a survey of other known E2F and SBF binding sites, and consider to what extent SBF could have taken over the sites. (More interesting would be to look at the binding sites in the extant species that are predicted to have both TFs and see if there's any evidence for both of them working. However, I understand that we're not supposed to suggest additional analyses at this stage.)

Reviewer #2:

A primary concern of the previous round of review was the distinction between E2F and SBF/MBF homologs in basal fungi. The new figure showing the alignment with strong conservation within each family is convincing to me that the distinction drawn is not caused by a limit in homology detection. Adding E2F family proteins to Figure 7 to illustrate the degree of divergence may be helpful, but I don't believe it is necessary given the statement that the HMMs never identified overlapping proteins or mis-identified known exemplars.

Overall, this work extends the earlier hypothesis that fungal SBF was transferred from viruses, by narrowing down the phylogenetic context of that transfer and identifying extant species with both SBF and E2F homologs. Much work must be done to characterize the degree to which those species truly have a "hybrid" network, but the present work lays an intriguing foundation. To me, the alternative hypothesis of rapid divergence between the SBF and E2F families is less parsimonious than horizontal transfer of SBF, although it is possible.

[Editors' note: further revisions were requested prior to acceptance, as described below.]

Thank you for submitting your article "Punctuated evolution and transitional hybrid network in an ancestral cell cycle of fungi" for consideration by *eLife*. Your article has been reviewed by two peer reviewers, and the evaluation has been overseen by a Reviewing Editor and Diethard Tautz as the Senior Editor.

The manuscript needs minor revisions.

Essential revisions:

Reduce the number of figures and make them less busy and higher quality (visible). Is each figure really essential? Can it be treated as a supplement to the primary figure (see how this is done for other papers we publish)? People are much less likely to read a paper with many busy figures.

Reviewer #1:

The authors have greatly improved the paper, and have addressed my concerns about technical issues related to the homology detection, as well as included phylogenetic analysis that shows the fungal proteins to be monophyletic, consistent with their model of a single transfer event. They have also added a reasonable bioinformatics analysis that shows the overlap in specificity between SBF and E2F.

Reviewer #2:

Overall, this work extends the existing viral-origin hypothesis for SBF, primarily by identifying species carrying both SBF and E2F. The additional analyses in this round of review further support this hypothesis. The authors more detailed analysis of HMM performance (Figure 6) gives more confidence in the bioinformatics, and the new phylogenetic analysis (Figure 8) makes the evolutionary hypothesis more specific. The new analysis potential binding of SBF to known human E2F binding sites also suggests that hijacking may have been possible (Figure 10), although it does not tell us much about what hijacked hybrid networks would look like.

At this point, I have no substantive concerns.

---

## [Author Response]

We thank you for the opportunity to revise and resubmit our manuscript.

We have modified the main text to address reviewer comments and we expanded details of data analysis in Methods. We also modified the Figures:

1) We reorganized the names and eukaryotic species tree in Figure 2 and Figure 3 to reflect the most up-to-date and modern eukaryotic classification (Adl et al., 2012).

2) We split old Figure 4 into new Figure 4 and new Figure 5.

3) We split old Figure 6 into new Figure 7 and new Figure 8.

4) We improved and updated new Figure 6 (E2F versus SBF sequence-structure) with a structure of fungal SBF/MBF bound to its MCB site (Liu et al., 2015).

5) We deleted old Figure 7 (fungal cyclin evolution) based on our reviewer feedback.

Summary:

*This paper addresses the evolution of cell cycle specific transcription factors and cyclins in fungi. All of us feel that the cyclin part was less interesting, more speculative, and poorly motivated. Your story about evolution of SBF-like transcription factors is the more interesting aspect. Unfortunately, the key discovery discussed in the paper, namely the origin of SBF from DNA viruses was reported long ago by Koonin and this aspect of the paper is therefore not entirely novel. More interesting is the finding that basal fungi appear to have both SBF and E2F/Rb like factors, confirming that E2F/Rb must have been replaced by SBF early on in the fungal lineage. In a way, one already knew that this must be case since the E2F system is present in plants as well as animals, implying that it must have been present in the common ancestor of fungi and animals, which are closer to each other than plants. We already knew therefore that E2F must therefore have been lost early in the fungal lineage. However, placing when this took place is certainly interesting and an important advance in our understanding.*

We thank the reviewers for recognizing the significance of our manuscript ("Punctuated evolution and transitional hybrid network in an ancestral cell cycle of fungi").

The Whi5-SBF pathway in yeast was only recently established as functionally analogous to the Rb-E2F pathway in mammals (Costanzo et al., 2004; de Bruin et al., 2004). These authors were careful to classify Whi5-SBF as "functional analogues" rather than homologs because Whi5-SBF lacked sequence identity to Rb-E2F. However, neither these authors nor the cell cycle community made the connection between viral KilA-N domains, SBF, hybrid networks, and early fungal evolution -- despite the fact that Koonin and colleagues had already published on the viral origins of APSES (Iyer et al., 2002).

Our manuscript is respectful of the existing literature. We cite multiple papers that already noted that Rb-E2F is likely ancestral because it is found in both plants and animals (Cao et al., 2010; Cross et al., 2011; Doonan and Kitsios, 2009; Fang et al., 2006; Harashima et al., 2013). However, none of these papers addressed the evolution of SBF versus E2F in the fungal lineage or established the example of cell cycle rewiring via a hybrid network. To this end, we think that our manuscript will be of great interest to the broader community that reads *eLife* (cell cycle, fungal cell biology, virology, and evolution).

*All of us were unimpressed by the experiment that E2F site from a histone promoter in mammals can replace the SBF sites in the CLN2 promoter. Results are also consistent with the obvious alternative hypothesis that SBF and E2Fs are distant homologues and the binding specificity has simply been conserved during evolution. We don't see how any experiment along these lines could be said to support the hijacking hypothesis (to the exclusion of the obvious competing hypothesis).*

We have rewritten our experimental section to better clarify the purpose of our experiments in budding yeast. By the time we motivate our experiments in subsection “SBF ancestor could regulate E2F-target genes”, we already established that SBF is not a distant homolog of E2F (new Figure 6) and that SBF (with homology to viral KilA-N domain) was derived by HGT (we cite work by Koonin & colleagues and our new Figure 7). Thus, the obvious alternative hypothesis that SBF and E2F are distant homologs, and that binding specificity has been conserved during evolution is highly unlikely.

Furthermore, even if SBF and E2F are non-homologous, they have overlapping binding specificity. Since sequence specific transcription factors allow degeneracy (or wobble positions) and bind to short DNA sequence families (motifs), these experiments show only that the E2F sites that were chosen probably overlap the specificity of SBF enough that they can be regulated. Therefore, these experiments really add nothing to the hijacking model, except to confirm the observation that the two transcription factors have overlapping specificity.

Briefly, the point of these experiments was to determine whether budding yeast SBF (a molecular "descendent" of the ancestral, viral-derived SBF) is able to recognize and bind a canonical E2F site. We clearly state that the known consensus motifs of SBF from budding yeast and E2F from animals have overlapping binding specificity, which is the observation that originally prompted our hijacking hypothesis. However, given that a single base pair difference can abrogate DNA binding, it was not obvious to us a priori whether yeast SBF would be able to bind a canonical E2F site. To our knowledge, no one has measured SBF binding to an E2F site. Hence, we felt obligated to test SBF binding to an E2F consensus site in vivo.

*Overall, the paper has one really key novel observation, namely the detection of the E2F/Rb system in basal fungi and this is certainly worth publishing. However, you need to show that there really are both classes of proteins in the basal fungi. You need to consider and rule out alternative explanations. For example, are you simply assigning some of the homologues in the basal fungi to be E2Fs and some to be SBF/MBFs because they have reached the limit of homology detections?*

We used Profile-Hidden Markov Models (profile-HMMs) to detect homologs for each of the families studied, using the HMMER 3 package (Eddy, 2011). Profile-HMMs are some of the most accurate and sensitive methods for detection of remote homologs (Eddy, 2011; Johnson et al., 2010). Starting with a set of diverse yet reliable protein homologs is fundamental for detecting remote protein homology and avoiding “model poisoning” (Johnson et al., 2010). To this end, we used reliable training-set homologs from the cell cycle model organisms *Arabidopsis thaliana*, Homo sapiens, *Schizosaccharomyces pombeS. pombe*, and *Saccharomyces cerevisiae*, to build the profile-HMMs used to detect E2F/DP and SBF/MBF homologs. All homologs recovered through profile-HMM search were further validated (or rejected) using an iterative search algorithm (Jackhmmer) against the annotated SwissProt database using the HMMER web server (Finn et al., 2011).

Our profile-HMM for E2F/DP family only detects E2F or DP, where as our profile-HMM for SBF/MBF family only detects SBF/MBF (or APSES). The same protein was never identified by both profile-HMMs because the sequence profiles and the structure are different; see discussion of E2F and SBF sequence-structure below. In the case of basal fungi, which have both E2F/DP and SBF/MBF, all proteins classified as an E2F/DP are clear homologs of E2F or DP (see alignment in new Figure 6) and all proteins that we classified as SBF/MBF are clear homologs of SBF/MBF (see alignment in new Figure 6). Thus, we have not misclassified E2Fs or SBF/MBFs in basal fungi and we have not reached the limit of homology detection. They are bona fide homologs.

We now elaborate on our identification of potential protein family homologs in the Methods section.

*The problem is that this key finding has been wrapped up in a paper that merely repeats Koonin's findings and addresses the evolution of fungal cyclins in a manner that is not particularly revealing. We would recommend a far shorter paper focusing on the one key finding.*

Based on this feedback, we have shortened the manuscript and removed old Figure 7. Our revised manuscript focuses on the evolution of cell cycle transcription factors and inhibitors (E2F-Rb and SBF-Whi5) and the hybrid network present in early diverging fungi.

We will publish our specific analysis of fungal cyclin evolution elsewhere.

*Reviewer 1:*

With regard to your models in Figure 7 and Figure 8, have you thought about alternative hypotheses? For example, CLN sets the stage for the acquisition of SBF. What evidences do you have to rule alternatives out?

We look forward to discussing this issue in our follow-up paper on fungal cyclin evolution.

*Reviewer 3:*

1) Figure 3 and Figure 4 are critical to the authors' argument, but the data are very poorly presented. The whole argument rests on the accurate detection of homologues for both types of regulators in the basal fungi. The authors need to take much more care to convince the reader that the observations are not simply due to errors in the homology detection (due for example to longer branches or denser sampling of species in the fungi). For example, they need to show convincing evidence that the diversity of the SBF/MBFs falls outside of the diversity of the E2Fs, and that the viral proteins fall within the SBFs and outside of the E2Fs. Figure 6a shows a phylogeny of SBFs, but it doesn't include the E2Fs. Even more important, they need to show that there really are both classes of proteins in the basal fungi. They need to consider and rule out alternative explanations. For example, are they simply assigning some of the homologues in the basal fungi to be E2Fs and some to be SBF/MBFs because they have reached the limit of homology detections?

As described above in our response to the summary critique, we are confident that we do not have errors in detection of homology in basal fungi.

*2) The structural evidence could represent key supporting data to show that E2Fs and SBFs really are two different families. However, is not presented in a very convincing way. "it is possible that fungal SBF is a duplicated and hyper-evolved E2F. However, the number and topology of secondary elements (i.e., α helices, β strands, and loops) that stabilize and pack around the core helix-turn-helix are different." Does the number and topology of secondary structural elements always stay the same in homologues? If not, how does this support the hypothesis that they are non-homologous? "Crystal structures of the DNA binding domains of E2F and SBF show that they are not structural homologues (Taylor* et al.*, 1997; Xu* et al.

, 1997) (see Figure 5)." Here the authors just state that E2F and SBF are not structural homologues. What is it about the structures in Figure 5 that shows that they are not homologues?

In response to the reviewer's questions, we have extended and clarified how the structural evidence supports the hypothesis that E2F/DP and SBF/MBF are two different families. We now write:

"The DNA-binding domains of SBF/MBF (Taylor et al., 2000; Xu et al., 1997) and E2F/DP (Zheng et al., 1999) are structurally classified as members of the winged-helix-turn-helix (wHTH) family, which is found in both prokaryotes and eukaryotes (Aravind and Koonin, 1999; Aravind et al., 2005; Gajiwala and Burley, 2000). Although the DNA-binding domains of E2F/DP and SBF/MBF are both classified as wHTH proteins, they show important differences in sequence, overall structure, and mode of protein-DNA complex formation that lead us to conclude that it is highly unlikely that they are orthologs.

The strongest arguments against SBF-E2F orthology are based on structural biology. Many wHTH transcription factors, including the E2F/DP family, have a ‘recognition helix’ that interacts with the major or minor grooves of the DNA. The E2F/DP family has an RRXYD DNA-recognition motif in its helix that is invariant within the E2F/DP family and is responsible for interacting with the conserved, core GCGC motif (Zheng et al., 1999) (see Figure 6: red structure). The RRXYD recognition motif is strikingly conserved in E2F/DP across all eukaryotes, including the E2F/DP proteins uncovered in basal fungi (Figure 6, left). In contrast, the first solved SBF/MBF structure, Mbp1 from *S. cerevisiae* in the absence of DNA, suggested Mbp1 recognizes its MCB (Mlu I cell cycle box, ACGCGT) binding site via a recognition helix (Taylor et al., 1997; Xu et al., 1997). A recent structure of PCG2, an SBF/MBF homolog in the rice blast fungus *Magnaporthe oryzae*, in complex with its MCB binding site does not support this proposed mode of DNA binding (Liu et al., 2015). In striking contrast to many wHTH structures, in which the recognition helix is the mediator of DNA binding specificity, the wing of PCG2 binds to the minor groove to recognize the MCB binding site. The two glutamines in the wing (Q82, Q89) are the key elements that recognize the core MCB binding motif CGCG (Figure 6, blue structure). Family-specific conservation in the DNA-binding domain is observed for all members of the SBF and APSES family, including basal fungal sequences (Figure 6, right). In summary, the incongruences in sequence, structure, and mode of DNA-interaction between E2F/DP and SBF/MBF families strongly suggest that SBF is not derived from E2F."

3) Even if the structural evidence is sound, and the proteins are non-homologous, the authors specific argument rests on the identification of both proteins in the basal fungi. The authors should show and explain the evidence that the putative homologues in the basal fungi are likely to adopt one structure and not the other. The alignment in Figure 5 shows many irrelevant species, and the homologues from the basal fungi are not highlighted. I think the authors need to consider their evolutionary model much more carefully and design experiments that actually test it.

We now highlight the E2F/DP proteins from basal fungi (bold font) in the sequence alignment of new Figure 6. We thank the reviewer for highlighting this point. We think revised Figure 6 makes it clear that E2F/DP and SBF/MBF are not homologs.

[Editors' note: further revisions were requested prior to acceptance, as described below.]

The reviewers see improvements in your revision but are still far from being satisfied. Ordinarily, this level of criticism on a revised submission would be considered unacceptable, however, we have decided to provide you with an opportunity to substantially revise the manuscript one last time. If you have questions about our instructions, please write and ask for clarification. Of course, given the concerns we have expressed we would understand if you choose to submit to a different journal. Please advise as to your course of action as soon as possible.

We thank you for giving us the opportunity to revise our manuscript. We include new data and analysis to address the Reviewer comments. Our new co-author, Dr. Raluca Gordân (Duke), has extensive experience generating and analyzing high-throughput in vitro DNA binding data of eukaryotic transcription factors, e.g. E2F (Afek et al., 2014). We modified the Figures as follows:

1) We now show the E-values of E2F (E2F_TDP.hmm) and SBF (KilA-N.hmm) HMM models against all predicted ORFS in our surveyed genomes (new Figure 6).

2) Old Figure 6 is new Figure 7.

3) Old Figure 7 is new Figure 8. We created a better phylogeny of aligned KilA-N domains with stronger confidence estimates on the deep branches between prokaryotic DNA viruses, eukaryotic DNA viruses, and the fungal SBF proteins.

4) Old Figure 8 is new Figure 9. We now include DNA binding motifs of human E2F (E2F1) and budding yeast SBF (Swi4) from the JASPAR database to motivate our preliminary hypothesis that SBF can bind E2F sites.

5) New Figure 10 analyzes the DNA binding specificity of human E2F1 and budding yeast SBF generated via high-throughput in vitro binding assays. We show there are E2F-only motifs, SBF-only motifs, and "common" motifs that can be bound by both E2F and SBF. We then scan human promoters (known to be regulated by E2F) with these in vitro E2F and SBF binding specificities to show that many human E2F binding sites could be bound by SBF.

6) Old Figure 9 is new Figure 11. We improved this figure to show possible roles of E2Fonly, SBF-only, and common DNA binding sites in the hybrid network of basal fungi.

*I encourage you to shorten this further (as per multiple reviewers' request).*

To address our Reviewer comments, the manuscript now includes new text, data, and figures. We moved towards more data and analysis rather than less. Given the level of skepticism from one of our Reviewers, we think it is important to show all the data and analysis to future readers. We note that our main text has 4950 words, which is below the suggested maximum of 5000.

Reviewer #1:

The authors did a good job of clarifying the structural data and present a much more convincing case that the two proteins are non-homologous. However, they did very little to address my other concerns. Rather than doing a detailed, focused analysis to convince the reader of their hypothesis that the basal fungi contain both transcription factors, and that the SBFs are inherited from a single horizontal transfer event, they simply bolded the basal fungi species names in an alignment. They did not show any phylogenetic analysis that would rule out competing evolutionary scenarios (such as recent transfer of the E2Fs into the basal fungi) or technical issues (such as homology detection problems). In response to my comments about the latter, they simply stated that their HMMs don't make any classification errors: "Our profile-HMM for E2F/DP family only detects E2F or DP, where as our profile-HMM for SBF/MBF family only detects SBF/MBF (or APSES)." In fact, I agree with the authors that the HMMs are probably very accurate, but why not show data to support this claim? (E.g., the score distributions or phylogenetic trees?) The authors don't seem to appreciate (or have chosen to ignore) the fact that remote homology detection (especially the claim that homologs are not present) is a statistical process and always has an element of uncertainty. (For example, there might be additional diverged SBF homologues in other eukaryotes that are below the E-value threshold, missed due to gene prediction errors, incomplete genome sequences, etc.) Since the detection of these homologs in the basal fungi and no SBFs in other eukaryotes is the key result of this paper, why not make a convincing case to support it (instead of the unnecessary Figure 2 and Figure 3)?

We address the Reviewer's key critiques below.

Critique #1: Convince readers of the hypothesis that the basal fungi contain both transcription factors (E2F and SBF), and convince readers that there have been no errors of remote homology detection or misclassification by HMM models.

To address our Reviewer's concern with homology detection, we plot the Pfam HMM E-values of E2F_TDP.hmm and KilA-N.hmm (the SBF DNA binding domain from Pfam) of all predicted proteins for any given genome in our new Figure 6. We used KilA-N.hmm for remote homology detection because pSMRC.hmm (our HMM model trained on fungal SBFs to detect SBF; see Figure 5—figure supplement 1) includes two well-conserved fungal ankyrin repeats in addition to the KilA-N DNA binding domain. In contrast, the KilA-N domain from Pfam was trained on a diverse set of KilA DNA-binding domains across bacterial DNA viruses, eukaryotic DNA viruses, and fungal SBF+APSES proteins. Thus, it should be a more sensitive HMM model to detect remote KilA-N homologues in other eukaryotic genomes. All initial hits with E-values between 1e-5 and 1e-10 were further validated (or rejected) using an iterative search algorithm (Jackhmmer) against the annotated SwissProt database using the HMMER web server (Finn et al., 2011). We then inspected these sequences manually for key conserved KilA-N residues.

Our controls (H. sapiens genome, Figure 6 and *S. cerevisiae* genome, Figure 6) demonstrate that E2F_TDP.hmm is specific to E2F/DP and that KilA-N.hmm is specific to SBF+APSES. We show that genomes with hybrid network (S. punctatus, Figure 6, and other basal fungi with both transcription factors, Figure 6) have both E2F/DP and SBF+APSES. E2F_TDP.hmm never hits an SBF+APSES transcription factor and KilAN. hmm never hits an E2F transcription factor (i.e. there are no scores on the diagonal of the panels in Figure 6). Thus, there is no misclassification by the Pfam HMM models. E2F and KilA-N share as much sequence identity with each other as they do with unrelated proteins. The conservation of known sequence-structure relationships in either E2F or SBF (new Figure 7) reinforces the fact that basal fungi have genuine E2Fs and SBFs.

Critique #2: Convince readers that SBFs were inherited from a single horizontal gene transfer event.

Pfam HMM E-values of E2F_TDP.hmm and KilA-N.hmm of our remaining eukaryotic genomes (Figure 6) show that we cannot detect SBF+APSES or other KilA-N domains outside of fungi (some initial hits with E-values in between 1e-5 and 1e-10 were subsequently shown to be false positives by an iterative Jackhmmer search on SwissProt database). However, as discussed in our manuscript and shown by Koonin's group (Iyer et al., 2002), fungal SBF DNA-binding domain shows homology to the KilA-N domain from prokaryotic and eukaryotic DNA viruses.

To broaden the scope of analysis beyond our downloaded eukaryotic genomes, we carefully surveyed all KilA-N domains detected by the Pfam database. The majority of known KilA-N domains (weighted by species, not the number of sequences) are found in prokaryotes (85%) with a smaller fraction (10%) found in eukaryotes and a smaller fraction found in DNA viruses (5%). The KilA-N domains in prokaryotes appear to be either integrated by or derived from prokaryotic DNA viruses (i.e. bacteriophage), and thus, we will treat them as such. Within the eukaryotes, all known KilA-N domains are found in fungal genomes with three notable exceptions.

The first exception is *Trichomonas vaginalis*, a parasitic excavate with 1000+ KilA-N domains (Figure 6). The *T. vaginalis* KilA-N domains have top blast hits to prokaryotic and eukaryotic DNA viruses, e.g. Mimivirus, a large double-stranded DNA virus of the Nucleo-Cytoplasmic Large DNA Viruses (Yutin et al., 2009). Mimiviruses are giant viruses known to infect simple eukaryotic hosts, such as *Acanthamoeba* and possibly other eukaryotes (Abrahão et al., 2014; Raoult and Forterre, 2008). The second and third exceptions are found in two insects, *Acyrthosiphon pisum* ('pea aphid') and *Rhodius prolixus* ('triatomid bug'). The one KilA-N domain in A. pisum genome has a top blast hit to eukaryotic DNA viruses (e.g. Invertebrate Iridescent Virus 6). The three KilA-N domains in R. prolixus have top blast hits to prokaryotic DNA viruses (e.g. Enterobacteria phage P1). The diverse and sparse distribution of KilA-N domains throughout the eukaryotic genomes is consistent with their horizontal gene transfer into hosts from eukaryotic DNA viruses and/or via engulfed bacteria that were infected with prokaryotic DNA viruses. In fact, the horizontal transfer of genes between Mimivirus and their eukaryotic host, or the prokaryotic parasites within the host, has been shown to be a more frequent event that previously thought (Moreira and Brochier-Armanet, 2008).

To gain further insight into the possible evolutionary origins of the SBF subfamily via horizontal gene transfer, we aligned diverse KilA-N sequences from the Uniprot and PFAM database to the KilA-N domain of our most basal fungal SBF+APSES sequences (Zoosporic fungi (“chytrids”) and “Zygomycetes”) and built a phylogenetic tree (new Figure 8). There are three major phylogenetic lineages of KilA-N domains: those found in eukaryotic viruses, prokaryotic viruses, and the fungal SBF+APSES family. Our results show that the fungal SBF family is monophyletic and strongly supported by multiple phylogenetic support metrics, which suggests a single HGT event as the most likely scenario that established the SBF+APSES family in a fungal ancestor. However, our current phylogeny is unable to distinguish whether the SBF family arrived in a fungal ancestor through a eukaryotic virus or a phage-infected bacterium. Structural and functional characterization of existing viral KilA-N domains could help distinguish between these two hypotheses.

Critique #3: Convince the reader that E2Fs were not horizontally transferred into basal fungi.

Figure 3—figure supplement 2 is a phylogenetic analysis of all eukaryotic E2Fs. Our phylogenetic tree shows that fungal E2Fs are adjacent to metazoan E2Fs, as expected from the known eukaryotic species tree. This phylogeny is consistent with the hypothesis that fungal and metazoan E2F were vertically inherited from an opisthokont ancestor, rather than fungi losing E2F and having them horizontally transferred from another organism.

We now include new Figure 6–Figure 8 and parts of this discussion in the main text.

*Similarly, they have not changed the presentation of the experiment demonstrating that the SBF can bind to an E2F binding site. In their response, they claim that this experiment is important because "a single base pair substitution in the SBF motif can reduce gene expression". However, they have still not showed which substitution that was. Therefore the argument is misleading: all substitutions in transcription factor binding sites are not equivalent, and this experiment would only be interesting if the substitution between the histone binding sites and the SBF consensus was one of those that was likely to reduce expression. If it is, why not show this and explain it clearly?*

Our statement that “a single base pair substitution in the SBF motif can reduce gene expression” was meant as a general statement. For many transcription factor binding sites, even changing a single base pair may result in decrease binding and subsequently decreased gene expression. In our experiment, the change in the SBF site was much more substantial than a single base pair mutation: we replaced the SBF sites (TCACGAAA) of CLN2 (Koch et al., 1996) with a known E2F binding site (GCGCGAAA) from the promoters of the histone cluster genes (Rabinovich et al., 2008). There are, of course, other possible E2F DNA binding sites that we could have used in our experiment; we picked this one because it is a well-characterized E2F binding site.

To circumvent this sampling bias, we now analyze data from high-throughput protein-binding microarray (PBM) assays (Afek et al., 2014; Badis et al., 2008) of human E2F (E2F1) and budding yeast SBF (Swi4). PBM assays measure, in a single experiment, the binding of recombinant proteins to tens of thousands of synthetic DNA sequences, guaranteed to cover all possible 10-bp DNA sequences in a maximally compact representation (each 10-mer occurs once and only once). We used these PBM data to generate DNA motifs for E2F and SBF (Berger et al., 2006), and to compute, for each possible 8-bp DNA sequence, an enrichment score (or E-score) that reflects the specificity of the protein for that 8-mer. E-scores vary between -0.5 and +0.5, with larger values corresponding to higher affinity binding sites (Berger et al., 2006). As shown in new Figure 10A, E2F1 and Swi4 can bind a set of common motifs. For example, the E2F binding site variant that we tested in budding yeast (GCGCGAAA, highlighted in red), is one of the sites commonly bound in vitro by E2F and SBF.

Most notably, the in vitro PBM data show that there are specific motifs that can be bound only by E2F or only by SBF. To identify the key nucleotide differences between E2F-only and SBF-only binding, we created motifs of E2F-only and SBF-only sites. The consensus E2F-only (NNSGCGSN) and SBF-only (NNCRCGNN) motifs indicate that differential specificity between E2F1 and SBF is mediated by the nucleotides in the 3rd and 4th positions (underlined). Thus, to address our Reviewer's comment, E2F has a strict preference for G in the 4th position, whereas SBF has a strict preference for C in the 3rd position (Figure 10A).

We now include new Figure 10 and this discussion in the main text.

*Furthermore, since E2Fs regulate many cell cycle target genes, it would be much more convincing if the authors simply surveyed the known E2F binding sites and estimated (bioinformatically) which of them SBF is likely to regulate. The observation that SBF can bind the histone binding site (on its own) really adds nothing.*

We scanned the promoters of known E2F target genes from the human genome (CCNE1, E2F1, EZH2) with our empirically-defined DNA binding sites from PBM assays (Afek et al., 2014; Badis et al., 2008) to predict putative E2F-only, SBF-only, and common sites (Figure 10B). As expected, there are many predicted E2F-only and common (E2F & SBF) sites that can be bound by E2F in these known target genes. However, we can also find many potential SBF-only binding sites in these promoters. We extended our analysis to 290 known E2F target genes in the human genome to test the generality of SBF cross-binding to E2F sites (Figure 10C). Most E2F target promoters could be bound by SBF, which supports the hijacking hypothesis where an ancestral SBF might have taken control of several E2F-regulated genes.

We now include new Figure 10 and this discussion in the main text.

We note that our observation that SBF can bind an E2F binding site in vivo, leading to cell cycle regulated gene expression, is critical for showing that our results based on high-throughput in vitro data are also true in a cellular (in vivo) environment.

*I think the most important thing for the authors to add is a phylogenetic analysis showing that the data support the hypothesis of a single transfer event in the ancestral fungi.*

Please see new text and Figure 8.

As a secondary point, I guess if they really want to include the histone binding site experiment, they should include it in the context of a survey of other known E2F and SBF binding sites, and consider to what extent SBF could have taken over the sites. (More interesting would be to look at the binding sites in the extant species that are predicted to have both TFs and see if there's any evidence for both of them working. However, I understand that we're not supposed to suggest additional analyses at this stage.)

Please see new text and Figure 10.

[Editors' note: further revisions were requested prior to acceptance, as described below.]

Essential revisions:

*Reduce the number of figures and make them less busy and higher quality (visible). Is each figure really essential? Can it be treated as a supplement to the primary figure (see how this is done for other papers we publish)? People are much less likely to read a paper with many busy figures.*

We removed 2 figures (for a total of 9 essential figures) and decreased the level of detail from one figure. When necessary, we uploaded figures of higher quality (500 dpi), such that all species/proteins in large phylogenies (e.g. Cyclins) can be viewed using magnify/zoom function.

Here’s a list of the major changes:

1) We changed old Figure 3 (detailed table) into new Figure 2—figure supplement 1.

2) We changed old Figure 5 (detailed table) into Figure 3—figure supplement 1.

3) We moved panels (B) and (C) from old Figure 10 into Figure 8—figure supplement 1 and Figure 8—figure supplement 2.